METHODS AND RESOURCES

# Single-cell transcription analysis of *Plasmodium vivax* blood-stage parasites identifies stage- and species-specific profiles of expression

**Juliana M. Sà**[1], **Matthew V. Cannon**[2], **Ramoncito L. Caleon**[1], **Thomas E. Wellems**[1], **David Serre**[2]*

**1** Laboratory of Malaria and Vector Research, National Institute of Allergy and Infectious Diseases, National Institutes of Health, Bethesda, Maryland, United States of America, **2** Institute for Genome Sciences, University of Maryland School of Medicine, Baltimore, Maryland, United States of America

* dserre@som.umaryland.edu

**Data Availability Statement:** All sequence data generated in this study are deposited at the Sequence Read Archive under the BioProject

## Abstract

*Plasmodium vivax* and *P. falciparum*, the parasites responsible for most human malaria worldwide, exhibit striking biological differences, which have important clinical consequences. Unfortunately, *P. vivax*, unlike *P. falciparum*, cannot be cultivated continuously in vitro, which limits our understanding of its biology and, consequently, our ability to effectively control vivax malaria. Here, we describe single-cell gene expression profiles of 9,215 *P. vivax* parasites from bloodstream infections of *Aotus* and *Saimiri* monkeys. Our results show that transcription of most *P. vivax* genes occurs during short periods of the intraerythrocytic cycle and that this pattern of gene expression is conserved in other *Plasmodium* species. However, we also identify a strikingly high proportion of species-specific transcripts in late schizonts, possibly associated with the specificity of erythrocyte invasion. Our findings provide new and robust markers of blood-stage parasites, including some that are specific to the elusive *P. vivax* male gametocytes, and will be useful for analyzing gene expression data from laboratory and field samples.

## Introduction

Following the bite of an infected mosquito, *Plasmodium* parasites in the form of sporozoites move from the skin to the circulation, before invading the host hepatocytes. There, they multiply in their individual host cells before exiting into the bloodstream, where they proliferate in continuous rounds of asexual multiplications within erythrocytes (or red blood cells [RBCs]). This intraerythrocytic life cycle that is completed every 2 days by the most prevalent human parasites, *Plasmodium falciparum* and *P. vivax*, leads to an exponential increase in the number of parasites and is responsible for the disease symptoms in malaria patients. During the blood-stage infection, individual parasites progress through the successive morphological stages of rings, trophozoites, and schizonts [1,2]. Alternatively, some erythrocytic parasites differentiate

PRJNA603327 (https://www.ncbi.nlm.nih.gov/bioproject/PRJNA603327).

**Funding:** This work was supported by an award from the National Institutes of Health to the University of Maryland School of Medicine (U19 AI110820) and by the Intramural Research Program of the National Institute of Allergy and Infectious Diseases, National Institutes of Health. The funders had no role in study design, data collection and analysis, decision to publish, or preparation of the manuscript.

**Competing interests:** The authors have declared that no competing interests exist.

**Abbreviations:** bp, base pairs; CRT, chloroquine resistance transporter; FDR, false discovery rate; GR, gametocyte ring; IG, immature gametocyte; MDR1, multidrug resistance gene 1; MG, mature gametocyte; MSP, merozoite surface protein; qRT-PCR, quantitative reverse transcription polymerase chain reaction; RBC, red blood cell; PCA, principal component analysis; scRNA-seq, single-cell RNA sequencing; UTR, untranslated region.

into sexual stages, the male and female gametocytes, that mature to infect mosquitoes, undergo fertilization, and produce sporozoites that can then be propagated to new vertebrate hosts [3,4].

Several studies have shown that the morphological changes occurring during the intraerythrocytic cycle of the *Plasmodium* parasites are accompanied by important transcriptional differences [5–10]. For human parasites, these studies have primarily relied on analysis of synchronized parasites from in vitro cultures, which might not always recapitulate accurately the gene expression profiles observed in vivo [11,12]. Furthermore, analyses of samples collected at different time points along the development of tightly synchronized cultures cannot inform on whether gradual or abrupt transcriptional changes accompany the morphological changes observed microscopically. The reliance on cultivated parasites has also greatly limited analyses of nonfalciparum *Plasmodium* species that cause malaria in humans. In particular, despite the great burden of vivax malaria worldwide [13,14] and pioneering work conducted using ex vivo cultures [15,16], few studies have investigated the gene expression profiles of *P. vivax* parasites [17–20]. This paucity of information is problematic given the deep evolutionary divergence of *P. vivax* and *P. falciparum* species and the profound differences in their biology, including distinct microscopic presentations of asexual and sexual stage parasites, different rates of gametocytogenesis [21–23], specific RBC preferences and underlying invasion mechanisms [24–27], and differences in host tissue affinities and sequestration mechanisms during blood-stage infections [28,29].

Advances in genomic technologies, and notably the development of tools enabling the characterization of the gene expression profiles from individual cells [30], are providing new approaches to disentangle the inherent heterogeneity of most blood-stage infections and to improve our understanding of the regulation of transcription in malaria parasites. Applications of single-cell RNA sequencing (scRNA-seq) technology to *P. falciparum* and *P. knowlesi* in vitro cultures and to the rodent parasite *P. berghei* have provided unique insights into the molecular mechanisms underlying gametocytogenesis [31,32] as well as the regulation of the asexual intraerythrocytic life cycle [33,34]. Unfortunately, the requirement for fresh and viable parasites has so far limited extension of such studies to other *Plasmodium* species. Here, we describe scRNA-seq data generated from 7 New World monkeys infected with 4 different strains of human *P. vivax* parasites. Our results highlight genes that can serve as markers for different developmental stages, including male or female gametocytes, provide insights on the possible role of many incompletely characterized protein-coding genes, and reveal unique features of the profiles of gene expression during the *P. vivax* intraerythrocytic cycle.

## Results and discussion

### scRNA-seq provides detailed gene expression information on individual blood-stage *P. vivax* parasites

We generated scRNA-seq data from 10 *P. vivax*–infected blood samples collected from 7 New World monkeys: 4 samples were collected from 4 splenectomized *Saimiri* monkeys, each infected with 1 of 3 *P. vivax* strains; 3 samples were collected from 3 splenectomized *Aotus* monkeys infected with 1 of 3 strains of *P. vivax* strains; and 3 additional blood samples were collected from the same *Aotus* monkeys 16 hours after administration of 1 dose of chloroquine (Table 1 and S1 Fig). Upon collection, we processed each blood sample using MACS columns [35] to enrich for infected RBCs and immediately prepared 3′-end scRNA-seq libraries using the 10X Genomics technology [30]. We then sequenced each library to generate, on average, 125.5 million paired-end reads of 75 base pairs (bp) per sample (Table 1). Between 41.2% and 84.2% of the reads from each sample mapped to the *P. vivax* P01 genome sequence [36], with

**Table 1. Description of the samples and summary of the transcriptome data.** The table shows information about the sample (*P. vivax* strain, host species and animal identifier, and collection time in days post infection), microscopy data (parasitemia in percentage of infected RBCs, number of rings, trophozoites, schizonts and (female) gametocytes per 10,000 RBCs (and in %)) and sequencing data (number of reads generated, percentage of reads mapped to the *P. vivax* genome, number of individual parasite transcriptomes obtained and number (and percentage) of asexual, female, and male parasites).

| Sample | | | Microscopy | | | | | Sequencing | | | | | |
|---|---|---|---|---|---|---|---|---|---|---|---|---|---|
| Strain | Host ID | Collection | Parasitemia | Rings | Trophozoites | Schizonts | Gametocytes | No. Reads | % *P. vivax* | Parasites | Asexuals | Females | Males |
| Chesson | *Aotus* 86436 | 12 DPI | 0.16% | 10 (62.5%) | 1 (6.3%) | 3 (18.8%) | 2 (12.5%) | 151,711,653 | 84.2% | 1,037 | 917 (88.4%) | 108 (10.4%) | 12 (1.2%) |
| Indonesia-I/CDC | *Aotus* 86574 | 12 DPI | 0.62% | 24 (38.7%) | 38 (61.3%) | 0 (0.0%) | 0 (0.0%) | 147,583,347 | 77.4% | 2,098 | 1,816 (86.6%) | 236 (11.2%) | 46 (2.2%) |
| AMRU-I | *Aotus* 86416 | 12 DPI | 0.34% | 3 (8.8%) | 23 (67.6%) | 4 (11.8%) | 4 (11.8%) | 139,578,955 | 78.9% | 1,709 | 1,403 (82.1%) | 295 (17.3%) | 11 (0.6%) |
| Chesson | *Aotus* 86436 | 16 DPI[a] | 0.46% | 23 (50.0%) | 12 (26.1%) | 9 (19.6%) | 2 (4.3%) | 87,132,416 | 73.1% | 521 | 273 (52.4%) | 238 (45.7%) | 10 (1.9%) |
| Indonesia-I/CDC | *Aotus* 86574 | 16 DPI[b] | 0.45% | 10 (22.2%) | 27 (60.0%) | 8 (17.8%) | 0 (0.0%) | 92,108,991 | 67.9% | 589 | 500 (84.9%) | 89 (15.1%) | 0 (0.0%) |
| AMRU-I | *Aotus* 86416 | 16 DPI[a] | 0.47% | 12 (25.5%) | 29 (61.7%) | 0 (0.0%) | 6 (12.8%) | 110,785,183 | 80.7% | 1,795 | 1,738 (96.8%) | 52 (2.9%) | 5 (0.3%) |
| Chesson | *Saimiri* 4215 | 24 DPI | 0.03% | 1 (33.3%) | 0 (0.0%) | 1 (33.3%) | 1 (33.3%) | 165,620,223 | 51.0% | 249 | 233 (93.6%) | 16 (6.4%) | 0 (0.0%) |
| AMRU-I | *Saimiri* 5107 | 24 DPI | 0.01% | 1 (100.0%) | 0 (0.0%) | 0 (0.0%) | 0 (0.0%) | 127,962,320 | 41.2% | 22 | 21 (95.5%) | 1 (4.5%) | 0 (0.0%) |
| NIH-1993 | *Saimiri* 3879 | 15 DPI | 0.24% | 4 (16.7%) | 18 (75.0%) | 0 (0.0%) | 2 (8.3%) | 125,218,128 | 70.5% | 928 | 518 (55.8%) | 401 (43.2%) | 9 (1.0%) |
| NIH-1993 | *Saimiri* 5541 | 21 DPI | 0.04% | 2 (50.0%) | 2 (50.0%) | 0 (0.0%) | 0 (0.0%) | 107,172,479 | 72.0% | 267 | 225 (84.3%) | 41 (15.4%) | 1 (0.4%) |

[a]16 hours after animal received a single 10 mg/kg chloroquine dose orally.

[b]16 hours after animal received a single 5 mg/kg chloroquine dose orally.

**Abbreviations:** DPI, days post infection; RBC, red blood cell

most of the remaining reads mapping to the hosts' genomes (S1 Table). Interestingly, after these reads were assigned into individual cells, the vast majority of the single-cell transcriptomes consisted of reads that mapped exclusively to either the host or *P. vivax* genome (S2 Fig). In fact, less than 5% of all 9,215 parasite transcriptomes that passed stringent quality cutoffs (see below) contained more than 300 unique reads mapping to the corresponding host genome. This observation was surprising given that *P. vivax* preferentially invades reticulocytes [37,38], which are defined by the presence of RNA molecules (that are later lost during RBC maturation [39,40]). However, this relative paucity of host sequences can be explained by the selection of relatively mature *P. vivax* stages during MACS column purification (see below) and by the findings of an ex vivo study, which demonstrated that *P. vivax* parasites rapidly remodel the reticulocytes and actively expulse the intracellular content as they develop [27]. Most of the noninfected monkey cells harbored high levels of hemoglobin mRNAs, indicating that these were likely erythrocytes or of platelet-specific transcripts (e.g., platelet-factor 4 or pro-platelet basic protein genes).

After removing PCR duplicates, very few transcriptomes (between 0 to 86 per sample, with a median of 8.5) contained more than 75,000 unique reads, and these were excluded because they might correspond to droplets containing multiple parasites or other technical artefacts. We next compared the results using either all *P. vivax* transcriptomes with more than 1,000 unique reads (*n* = 13,503) or those with more than 5,000 reads (*n* = 9,215). Overall, the results were qualitatively equivalent (S3 Fig versus Fig 1), with only a slight increase in the proportion

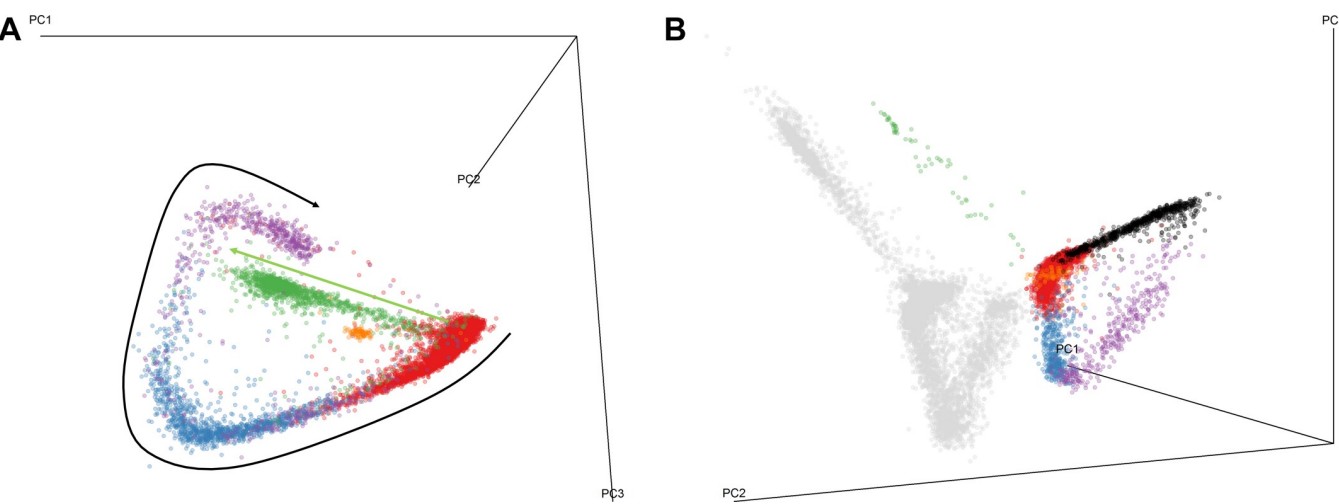

**Fig 1. PCAs of the individual *P. vivax* transcriptomes.** (A) PCA of 9,215 *P. vivax* parasites using 13,054 windows of 500 bp with at least 0.25% of the reads of one parasite. Each dot represents a single parasite transcriptome and is colored based on the expression of 60S ribosomal protein L34 (red), AMA1 (blue), MSP3 (purple), Pvs25 (green), and MGET (orange). The arrows represent the trajectories used to calculate developmental pseudotimes for asexual (in black) and female gametocytes (in green). (B) PCA calculated jointly from the data of 9,215 *P. vivax* parasites and 4,884 *P. berghei* parasites using the expression levels of orthologous genes. The gray dots represent *P. vivax* parasites and the colored dots show *P. berghei* parasites according to their developmental stages (yellow, rings; red, trophozoites; blue, schizonts; green, female gametocytes; orange, male gametocytes). Underlying data plotted in panels A and B are provided in S1 Data. bp, base pairs; PCA, principal component analysis.

of late schizonts excluded by the more stringent cutoff (S4 Fig). We chose to concentrate for the remainder of the analyses on the 9,215 parasites with at least 5,000 reads because these data provided a deeper characterization of the *P. vivax* transcriptomes (Table 1). On average, each of the individual parasite transcriptomes consisted of 19,282 unique sequences.

Because the 10X 3'-end assay relies on oligo-dT capture and priming and fragmentation of the cDNA into short molecules before sequencing, all scRNA-seq reads should derive from the last approximately 300 bp of each polyadenylated mRNA molecule [30]. Indeed, 68% to 72% of the *Aotus* reads mapped within 500 bp of the 3'-end of annotated genes (on either side). By contrast, only 42% to 48% of the reads mapped to the *P. vivax* P01 genome were located within 500 bp of one annotated gene 3'-end (see also S5 Fig), reflecting (i) the incomplete annotation of untranslated regions (UTRs) in this species [18] and/or (ii) the presence of unannotated or misannotated *P. vivax* transcripts. To further examine whether the scRNA-seq reads correspond to genuine signals from misannotated and unannotated transcripts, we compared the locations of the mapped reads with the 3'-end of transcripts predicted from bulk RNA-seq data from one *P. vivax*–infected patient (see Methods). Despite the crudeness of these de novo gene predictions, 22% of the scRNA-seq reads mapped to the last 500 bp of the predicted transcripts. This figure corresponds a 3.4-fold enrichment over what would be expected solely by chance (by comparison, the enrichment in the 3'-end of current gene annotations was only 2.5-fold) and highlights that the scRNA-seq reads likely represent many transcripts incompletely described in the current annotations. Overall, 45% of the scRNA-seq reads mapped to the last 500 bp of the 3'-end of either an annotated or predicted transcript. To circumvent the incomplete annotations of *P. vivax* genes, we summarized the scRNA-seq data, not by annotated genes but by nonoverlapping windows of 500 bp (which corresponds roughly to twice the width of the scRNA-seq peaks; see, e.g., S6 Fig), and we annotated these windows only afterward, with regard to their distance to the 3'-end of the closest gene on the same strand (see S2 Fig and Methods), with the caveat that the signal might actually derive from another unannotated transcript. Overall, we detected expression of 1,079 genes on average in each

individual *P. vivax* parasite; although, the majority of these genes was represented by very few reads, as typically observed with this technology (S2 Table).

## Multiple developmental stages are simultaneously present in each infection

Although certain *P. vivax* stages have been reported to accumulate in host tissues such, as bone marrow [19,41], the patterns, extent, and mechanisms of sequestration differ substantially from those of *P. falciparum*. As a consequence, and in contrast with *P. falciparum* infections, in most *P. vivax* infections, many asexual and sexual stages can be found simultaneously in the circulation [29,37]. Consistent with this reduced sequestration, principal component analysis (PCA) of the individual *P. vivax* transcriptomes revealed distinct populations of blood-stage parasites. Two of the first 3 principal components defined the main population of parasites, organized according to a shape reminiscent of a cycle, though with a conspicuous gap (Fig 1A and S7 Fig). Parasites at one end of this distribution expressed primarily genes associated with transcription and translation, whereas the parasites at the other end expressed many invasion protein genes (Figs 1A and 2A), suggesting that this population of parasites represented asexual blood-stage parasites progressing in development from trophozoites to schizonts. Note that, because they contain little hemozoin, ring-stage parasites are typically lost during MACS column purification, which likely explains their absence from our data and the gap in the cycle. The second principal component separated 2 additional populations of parasites from the main group: one population expressed known female gametocyte genes (e.g., Pv25 [42]), whereas a smaller population expressed genes known to be specific to male gametocytes (e.g., MGET [42]; Fig 1A).

To further examine these cell populations, we compared the scRNA-seq profiles of *P. vivax* parasites with those generated using the same technology from *P. berghei* blood-stage parasites [34]. Despite the deep divergence between these 2 species and the substantial differences between their life cycles (including a 24-hour intraerythrocytic cycle in *P. berghei* compared to a 48-hour in *P. vivax*), the PCA profiles were remarkably similar (Fig 1B and S8 Fig). This observation was consistent with previous findings from microarray [7,15] and scRNA-seq [34] studies that demonstrated global conservation of the gene expression changes during the intraerythrocytic cycle of *Plasmodium* species. These results also corroborated our interpretation of the different cell populations and confirmed that rings were missing from our data set.

*P. vivax* parasites from different stages were observed in most monkey infections, but not all stages were necessarily observed in each sample, and their relative proportion varied extensively among samples (Table 1 and S9 Fig). These results were consistent with the observation that most *P. vivax* infections are asynchronous, although this high diversity in stages might have possibly been exacerbated in our data by the splenectomy of the monkeys that was used to avoid parasite clearance and the dangers of splenomegaly [43]. Note that analysis of allelic variants across nucleotides sequenced at high coverage in all samples provided no evidence of genetic heterogeneity in the individual infections (S1 Text), indicating that the different parasite stages observed in each infection were unlikely to represent mixed populations of *P. vivax* clones out-of-phase in their cycles.

## scRNA-seq provides robust markers of *P. vivax* male and female gametocytes

*P. vivax* and *P. falciparum* are distinguished by important features of their gametocytogenesis: *P. vivax* gametocytes appear earlier than *P. falciparum* gametocytes in blood-stage infections and display very different morphology than *P. falciparum* gametocytes [21,22]. Unfortunately, the lack of in vitro culture methods for *P. vivax* has drastically limited studies of sexual

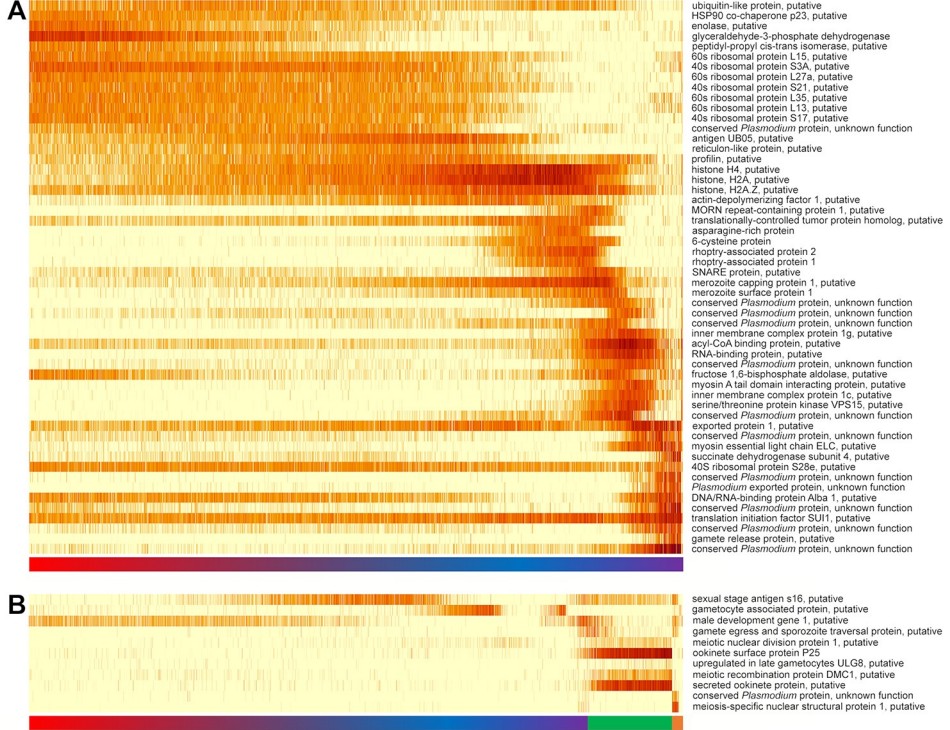

**Fig 2. Heatmaps showing variations in gene expression among blood-stage *P. vivax* parasites.** (A) Variations in the expression levels of the genes most expressed at different times of the *P. vivax* asexual development. (B) Variations in the expression levels of genes previously associated with gametocytes or gametes. Each row represents a different gene, and each vertical bar an individual *P. vivax* parasite ranked along the x-axis based on its pseudotime. The color of each bar represents the expression level of a specific transcript from nondetected (white) to highest expression (in red) in yellow-to-red scale. The color bar under the heatmaps shows the assignment of the parasites to different morphological stages (red, trophozoites; blue and purple, schizonts; green, female gametocytes; orange, male gametocytes). Underlying data plotted in panels A and B are provided in S2 Data.

commitment and gametocyte differentiation, and has also hindered the characterization of robust biomarkers for male and female *P. vivax* gametocytes. In this regard, the scRNA-seq data provided a unique opportunity to assess the expression of putative gametocyte markers and to identify novel candidates. Many of the genes typically expressed in *P. falciparum* or rodent gametocytes (Pvs25 [42], MGET [42], ULG8 [10], GEST [44]) were detected, at various levels, in male or female *P. vivax* gametocytes (Fig 2B). However, other genes labelled as "gamete" or "gametocyte" genes displayed patterns of expression in *P. vivax* that were not consistent with such annotations: for example, the "gametocyte-associated protein" (PVP01_1403000) or the "sexual antigen" s16 (PVP01_0305600) were mostly detectable in asexual parasites (Fig 2B). Importantly, transcripts of both of these genes were not detected together in individual asexual parasites and were therefore unlikely to indicate sexually committed parasites in the asexual population. In contrast, the patterns of expression of some genes were consistent with their transcription in differentiating gametocytes: the gamete egress and sporozoite traversal protein (GEST, PVP01_1258000) was detected only in the population of cells intermediate between the asexual parasites and the fully differentiated females, whereas the male development gene 1 (MDG1, PVP01_1435300) was expressed in both male and female putatively differentiating gametocytes (Fig 2B). To test more systematically whether our data set included

both immature and mature gametocytes, we examined the expression of genes previously associated with various stages of gametocyte differentiation [19]. Although the expression levels of genes associated with immature gametocytes were relatively constant along the female gametocyte pseudotime, we observed that the expression of mature gametocyte genes increased with developmental pseudotime (S10 Fig). In contrast, we did not observe any population of parasites expressing genes specifically associated with gametocyte rings, which could indicate (i) their loss during MACS column purification, (ii) insufficient sensitivity of the scRNA-seq, or (iii) their absence from circulating blood (e.g., due to sequestration in the bone marrow [19]). Interestingly, male gametocytes did not show elevated expression of genes previously associated with mature relative to immature gametocytes [19] (S10 Fig), suggesting that these markers may only be representative of female gametocyte differentiation, or that our data set only included immature but not mature male gametocytes.

We next searched the scRNA-seq data to identify genes highly expressed in male and/or female gametocytes and with little expression in other stages to generate a comprehensive list of putative biomarkers for male and female gametocytes (S3 Table). Interestingly, we noted that most of the genes highly expressed in female gametocytes (19 out of 25) were specifically expressed at this stage, whereas the genes highly expressed in male gametocytes were often also expressed at a different stage of the parasite life cycle, and these patterns were conserved in *P. berghei* (S3 Table). These patterns of sex-specific expression differ from the patterns of expression, at those same genes, measured previously by microarrays in *P. berghei* [45] and could reflect differences in regulation between *Plasmodium* species (though scRNA-seq data from *P. berghei* [34] are consistent with our findings from *P. vivax*) and/or differences in sensitivity and specificity of the methods used. On the other hand, male gametocytes appeared to express more species-specific genes (i.e., genes without orthologs in *P. falciparum* or *P. berghei*), possibly indicating differential evolutionary constrains on these different developmental stages (see also below).

Next, we reanalyzed the gene expression profiles previously obtained by bulk RNA-seq from whole blood of vivax malaria patients in Cambodia [20] using the transcripts deemed from the scRNA-seq data to be present exclusively in male or female *P. vivax* gametocytes. We determined that the expression of male and female gametocyte-specific transcripts did not significantly covary across patient infections (S11 Fig), confirming that the ratio of male/female gametocytes can differ greatly among patient infections [20]. These findings were also consistent with the variable proportions of male and female gametocytes detected by scRNA-seq in each New World monkey infection (Table 1).

## Gene expression is highly stage-specific and changes gradually during the *P. vivax* intraerythrocytic life cycle

Gene expression data from individual parasites revealed the exquisite regulation of *P. vivax* transcription along the intraerythrocytic life cycle: for the large majority of genes, expression was tightly restricted to a specific developmental stage, and the transcripts were not detectable elsewhere in the cycle (see, e.g., Fig 2A). This pattern was observed for members of the 6-cysteine gene family (Fig 3) as previously described [46] and held true for numerous other genes involved in metabolism, cellular processes, or cellular organization, reflecting the complex changes in parasite biology occurring as it grows and divides in the host RBC. The levels of histone expression also varied along the *P. vivax* intraerythrocytic cycle, recapitulating the patterns described previously [47]: for the genes encoding core histones (i.e., H2A, H2B, H3, and H4), expression increased gradually during asexual development and peaked in early schizont stages, consistent with their role in nucleosome assembly during replication, before decreasing

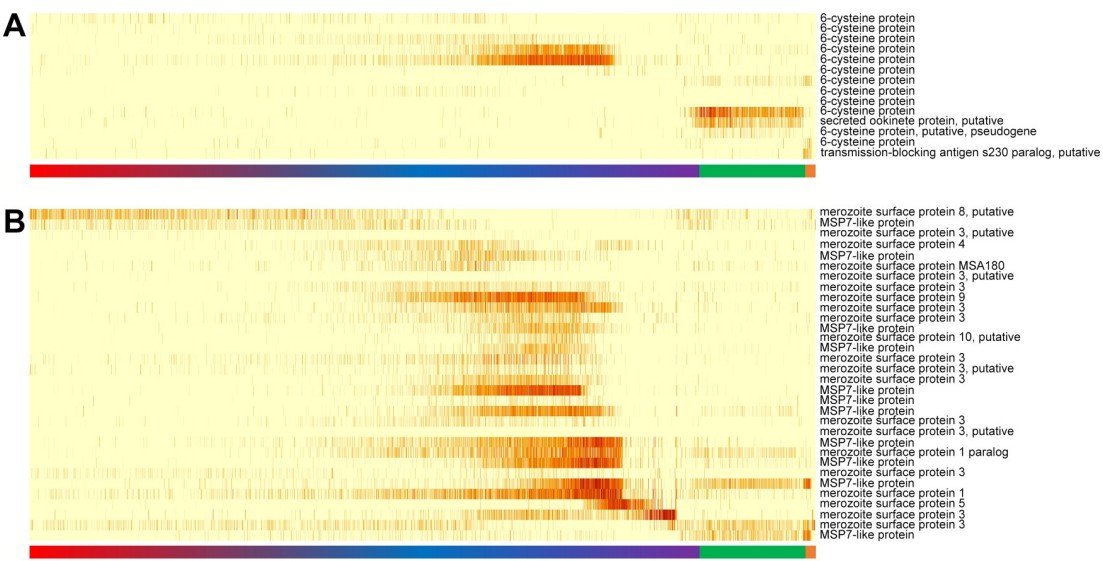

**Fig 3. Expression of genes from multigene families.** (A) Heatmap showing variations in the expression levels of 6-cysteine genes. (B) Heatmap showing variations in the expression levels of MSP genes. See legend of Fig 2 for details. Underlying data plotted in panels A and B are provided in S3 Data. MSP, merozoite surface protein.

below detectable levels later on (Fig 4A). By contrast, variant histones displayed relatively constant gene expression throughout the intraerythrocytic life cycle (e.g., H3.3) or had a delayed expression profile (H2Bv). Interestingly, these changes of histone expression were accompanied by changes in the expression of histone-regulating factors: the expression of the histone chaperone ASF1 that plays a key role in disassembly and reassembly of the chromatin during replication [48] peaked slightly earlier than the core histone expression, whereas the expression of the histone hairpin-binding protein that regulates histone mRNA processing in higher eukaryotes [49,50] preceded the increase in core histone mRNA levels (Fig 4B).

We also observed this stage-specific expression at genes commonly used for rapid diagnostic tests of *P. vivax* (S12 Fig). Although we cannot predict from the scRNA-seq data how the corresponding protein levels may vary across developmental stages, these observations suggested that detection of *P. vivax* parasites using aldolase [51,52] or lactate dehydrogenase [53] might be influenced by the predominant stages present in one infection. We therefore searched for genes that were highly expressed in most life cycle stages and might serve for diagnostics as well as for the normalization of transcript levels in quantitative reverse transcription polymerase chain reaction (qRT-PCR) experiments. We only identified a few genes highly expressed across the entire intraerythrocytic life cycle, including profilin (PVP01_0730900, S12 Fig), highlighting that the normalization of qRT-PCR assays and the correction for differences in stage composition will remain difficult challenges and continue to warrant careful evaluation of gene expression studies.

This tightly regulated, stage-specific gene expression was also observed for AP2 domain transcription factor genes. Despite their low expression levels, several AP2 genes could be reliably detected in our data set, and their expression was largely confined to specific developmental stages, with little overlap between them (S13 Fig). These observations corroborated the findings of studies conducted with other *Plasmodium* species and were consistent with the regulation of the entire intraerythrocytic cycle by successive switches in the expression of these master transcriptional regulators [15,54–56].

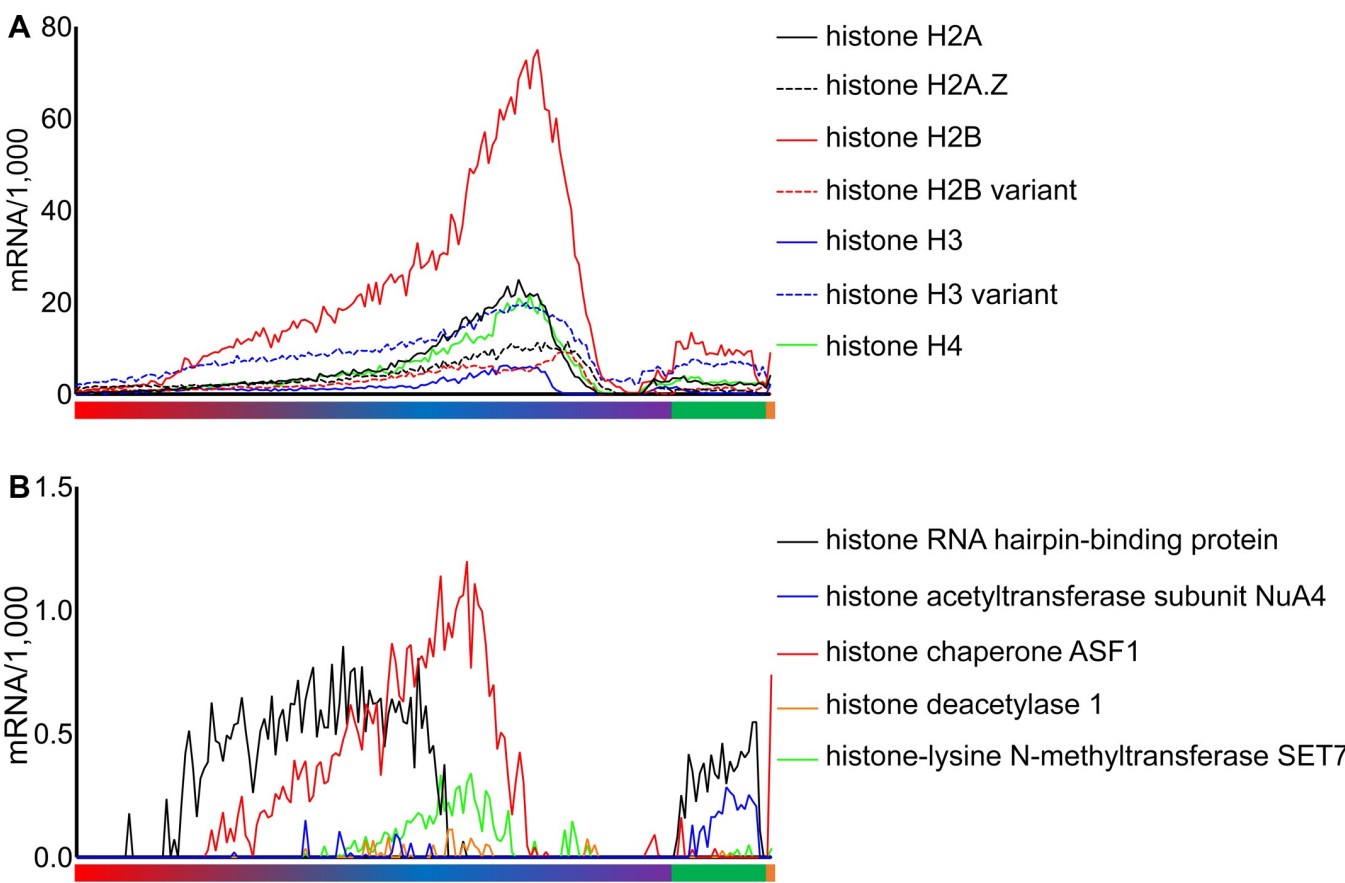

**Fig 4. Expression of histone and histone-related genes.** (A) Expression of histone genes according to the *P. vivax* development. The core histones are indicated by solid lines, and the histone variants are shown with dashed lines. Note that the expression level of core histones peaks in early schizonts, consistent with the timing of DNA replication and the need for new nucleosomes. (B) Expression of the 4 detectable histone-related genes according to the parasite development. Note how the histone RNA hairpin-binding protein (PVP01_1014900, in black) expression increases before replication, consistent with its role in nucleosome disassembly and reassembly, whereas ASF1 (PVP01_1442700, in red) peaks slightly before the increase in core histone, consistent with its role in histone mRNA regulation. The color bars under the graphs show the assignment of the parasites to different morphological stages (red, trophozoites; blue and purple, schizonts; green, female gametocytes; orange, male gametocytes). Underlying data plotted in panels A and B are provided in S4 Data.

Much of our knowledge about the changes in gene expression occurring during the *Plasmodium* intraerythrocytic cycle derives from microarray and bulk RNA-seq studies conducted on parasites cultured in vitro and sampled at discrete time points of their development. Such studies highlight differences in gene expression between parasites at different stages of their development, but it remained unclear whether these dynamic changes happened abruptly, mirroring the changes observed microscopically, or whether they occurred more continuously with development. Recent scRNA-seq data [33] indicated that most genes could be grouped into a few clusters with simultaneous changes in expression, supporting the hypothesis that gene expression changed abruptly in the intraerythrocytic life cycle of *Plasmodium* parasites. Our findings suggested gradual changes of gene expression resulting from continuous, rather than abrupt, switches in transcription (e.g., Figs 2A and 4). The hypothesis that gene expression changes occurred gradually was also supported by the continuous distribution of the parasites in the PCA plot (Fig 1), whereas distinct clusters of parasites would have been expected from discrete and coordinated regulation of gene expressions (note that this continuous distribution of asexual parasites is also present in other *Plasmodium* scRNA-seq data generated so

far [31–34]). One possible explanation for the discrete patterns described previously is that the parasites analyzed were not evenly distributed along the intraerythrocytic cycle, introducing apparent discontinuities, but additional studies will be needed to rigorously analyze and resolve the dynamics of these gene expression changes.

In addition to the transcription of specific genes at different stages of the intraerythrocytic development, the precise information generated by scRNA-seq from the 3'-end of genes also hinted at more subtle forms of gene expression regulation. For example, our data suggested that PVP01_0941100 may be transcribed with different 3'-UTRs in asexual and sexual parasites (S14 Fig). Combined with the observation that many reads mapped outside current gene annotations, the findings emphasized how incomplete our current understanding of the regulation of *P. vivax* gene expression is.

## Sequence homology in gene families does not necessarily predict patterns of expression or function

The *P. vivax* genome contains a number of multigene families defined by sequence homology and conservation of specific amino acid domains. Although the various members of these multigene families often have no known function, the role of a few families have been annotated based on the validated functions of particular proteins (e.g., the merozoite surface proteins [MSPs] [57]) or on their homologies to *P. falciparum* genes (e.g., the vir/PIR gene family [58,59]). Single-cell profiling provided a unique opportunity to test whether members of individual families were expressed at the same developmental stage, or in contrary, whether members of a family might be transcribed at different times and with different roles across the parasite life cycle. Although most MSPs were expressed late in the development of asexual parasites (Fig 3B), consistent with their role in erythrocyte invasion, 2 MSP genes displayed distinct expression patterns. An MSP3 gene (PVP01_1031100) was exclusively expressed in very late schizonts, in which it accounted for approximately 2.5% of all mRNA molecules. More surprisingly, an MSP7-like gene (PVP01_1219900) was expressed almost exclusively in gametocytes and at particularly high levels in male gametocytes (Fig 3B). The expression of MSP7-like genes was especially intriguing as 12 of these genes, clustered in 30 kb on chromosome 12, showed highly variable but tightly regulated expression in different stages (S15 Fig). Such differential transcription is consistent with various roles for different MSP genes at specific points in the *P. vivax* life cycle. Similarly, we observed vastly different expression profiles among members of the PHIST (S16 Fig), PIR (S17 Fig), and tryptophan-rich protein gene families (S18 Fig), suggesting that, despite their sequence homology, members of these individual families might not all have the same function.

## Schizonts display species-specific profiles of gene expression

Most *P. vivax* genes remain functionally unannotated (i.e., they are described as "conserved protein of unknown function"), and those that are annotated often carry a function determined from studies of their *P. falciparum* homologs. However, *P. falciparum* and *P. vivax* diverged from each other millions of years ago [60] and have very different biological features. Mirroring this divergence, 1,473 of the 6,276 annotated *P. vivax* genes (based on PlasmoDB-37 annotations) do not have annotated orthologs in the *P. falciparum* 3D7 genome. On average, 9.3% of the mRNA molecules in each *P. vivax* parasite were transcribed from annotated *P. vivax* genes without a *P. falciparum* ortholog. However, this pattern was not constant across all parasite stages and, in particular, late schizonts showed a pronounced difference, with up to 63.4% of the mRNA molecules derived from genes absent in *P. falciparum* (Fig 5). This remarkable result mirrored the important differences in the molecular mechanisms of human

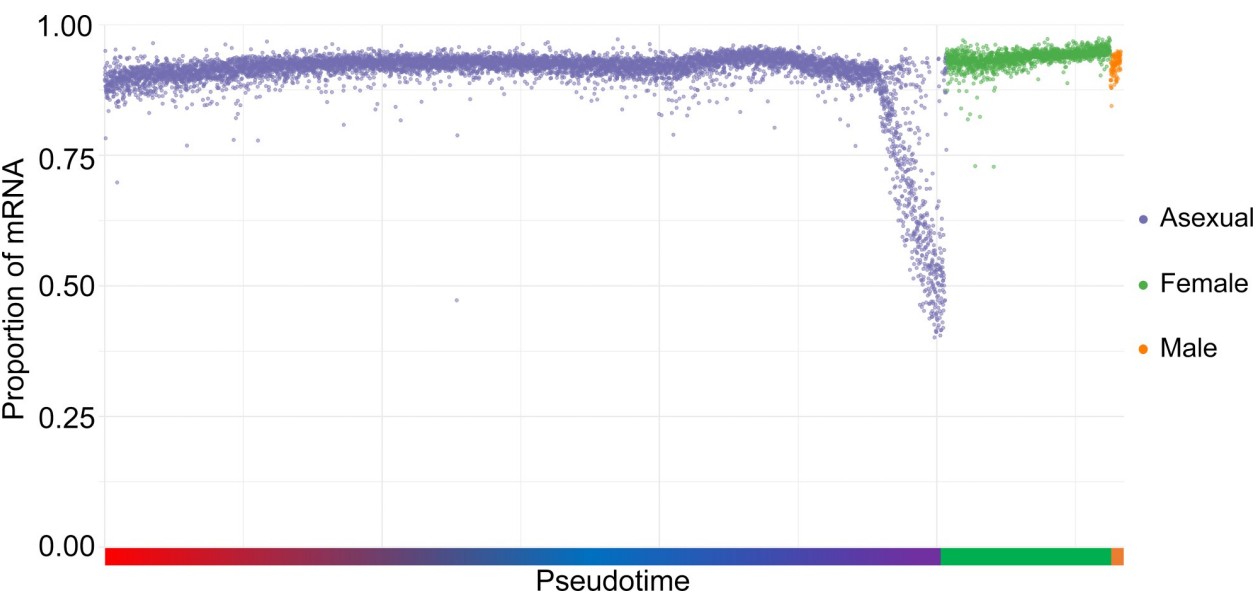

**Fig 5. Proportion of mRNA transcribed from *P. vivax* genes with a *P. falciparum* orthologs.** The figure shows the proportion of the mRNA molecules derived from *P. vivax* genes with at least one *P. falciparum* ortholog (y-axis) according to the parasite developmental stage (x-axis). Blue dots indicate asexual parasites organized according to their developmental pseudotime rank (from trophozoites on the left to schizonts on the right); green dots represent female gametocytes and orange dots male gametocytes. Underlying data are provided in S5 Data.

RBC invasion by *P. vivax* and *P. falciparum* [26] and highlight the limitations of *P. falciparum* as a model to understand *P. vivax* biology. Interestingly, this pattern was also observed when comparing *P. vivax* to *P. berghei* (S19 Fig), suggesting that the pathways underlying RBC invasion diverged faster than those regulating other features of the intraerythrocytic cycle. Interestingly, this analysis also confirmed that, overall, male gametocytes expressed a higher proportion of species-specific transcripts than female gametocytes.

## Influence of chloroquine treatment and host species on *P. vivax* gene expression

To statistically determine whether specific transcripts were differentially regulated upon chloroquine exposure, we compared the gene expression profiles of parasites from the same *Aotus* infection before and 16 hours after treatment, adjusting for the different distributions of developmental stages (see Methods). A total of 58 windows of 500 bp showed evidence of differential expression (false discovery rate [FDR] < 0.1) in the same direction and in all 3 infections (S4 Table). Interestingly, the genes encoding the core histones H2A, H3, and H4 were all significantly up-regulated after chloroquine exposure (see, e.g., S20A Fig). This finding might reflect the ability of chloroquine to intercalate into DNA and displace *Plasmodium* histones in vitro [61] and suggested that increasing histone abundance might be a mechanism of parasite response to drug exposure. In these short-term experiments, neither the chloroquine resistance transporter (CRT, PVP01_0109300) nor multidrug resistance gene 1 (MDR1, PVP01_1010900) showed significant changes in expression upon chloroquine exposure (uncorrected $p > 0.03$ and $p > 0.003$, respectively). Note, however, that the dose of chloroquine administered was subtherapeutic, and the gene expression changes reported here thus might not be particularly relevant to the biological processes underlying chloroquine-induced clearance or drug resistance.

We also tested for differences in gene expression that might be associated with the host species. *Aotus* and *Saimiri* erythrocytes differ in their ability to bind the invasion-related ligand Duffy binding protein 1 and are therefore likely to be invaded by *P. vivax* parasites using different molecular pathways [62,63]. Our data in *Aotus* were obtained from infections with 3 *P. vivax* strains—Chesson, Indonesia-1, and AMRU-1—whereas the data from *Saimiri* monkeys were mostly derived from infections of NIH-1993 parasites, with some contributions from Chesson and AMRU-1 infections (Table 1 and S9 Fig). This confounding effect precluded drawing firm conclusions about whether the differences in parasite gene expression were due to a response to the host rather than genetic polymorphisms between the parasites. Nevertheless, our analysis revealed potentially interesting genes differentially expressed between the *Aotus* and *Saimiri* infections: a number of these candidate genes expressed in late schizonts encoded exported proteins or MSPs (S5 Table), including a MSP3 gene that displayed a 2.1-fold overexpression in *Saimiri* (PVP01_1031700, S20B Fig). Our findings only marginally overlap with those of a recent whole-blood RNA-seq comparison of *Aotus* and *Saimiri* monkeys infected with the Salvador-I strain of *P. vivax* [62]: the MSP3 gene identified in our study was not orthologous to any of those identified previously, and the most differentially expressed tryptophan-rich antigen gene in schizonts only ranked 250th in our analyses (PVP01_0000140). These discrepancies could be due to the strain heterogeneity in our analyses or reflect the difficulty to adequately correct bulk RNA-seq analyses for even modest differences in stage composition.

Intriguingly, no expression of the DBP2 gene (PVP01_0102300) was detected in any of the samples analyzed here. Because we previously showed that this gene was deleted in the Salvador-I strain [64], we looked for evidence of gene expression of the neighboring genes. In all samples from the 4 *P. vivax* strains independently adapted to the monkeys, the region containing DBP2 and the upstream telomeric sequence (including 22 annotated genes) was devoid of any reads (aside from possibly mismapped sequences derived from multigene families). This observation could be consistent with the deletion of this region in all monkey-adapted *P. vivax* strains as we described in the Salvador-I strain. Interestingly, the boundaries of this putative deletion appeared different in the various strains, suggesting independent events, but always occurred immediately upstream of DBP2 or its immediate neighbor (S21 Fig). This fascinating finding could suggest that DBP2 is not necessary for invasion of *Aotus* or *Saimiri* monkey RBCs and may even be deleterious in these monkey models, leading to its systematic deletion. This contrasts with the frequent presence of the DBP2 gene in clinical *P. vivax* isolate genomes from around the world [65,66]. Although more studies will be required to rigorously understand the effect of the host species on parasite gene expression, our analyses suggest that scRNA-seq, using additional strains from monkey infections and/or parasites from patients with different RBC genotypes, will provide a useful framework to identify proteins involved in different pathways of RBC invasion.

## Conclusion

The results of our study highlighted the complex and dynamic changes in gene expression occurring during the intraerythrocytic cycle of *P. vivax* parasites. The vast majority of *P. vivax* genes appeared to be expressed only during a short developmental period and very few genes (if any) were expressed at the same level throughout the intraerythrocytic cycle. This paucity of constantly expressed "housekeeping" genes will complicate the interpretation of RT-PCR or bulk RNA-seq experiments, which will likely require several markers to adequately correct for the stage composition of the samples. The results of this study can provide a framework to implement such analyses by identifying genes specifically expressed at different stages of the

parasite development, including in male and female *P. vivax* gametocytes. Our study also highlighted that many *P. vivax* transcripts are incompletely annotated which complicates assigning the scRNA-seq signals to specific genes. Combining the single-cell gene expression profiling with long read sequencing may provide an opportunity to complete the annotation of *P. vivax* gene structures as well as to characterize different isoforms that may be expressed by the same genes. Similarly, the characterization of when specific genes are expressed in the parasite life cycle provides an opportunity to functionally annotate these genes that often have no predicted function. Finally, the scRNA-seq data revealed differences in the regulation of *P. vivax* compared to *P. falciparum* parasites, particularly at the schizont stage, underscoring the importance of specific studies directed to each of these major human pathogens.

## Methods

### Ethics statement

All animal procedures were conducted in accordance with the National Institutes of Health (NIH) guidelines and regulations [67], under approved protocols by the National Institute of Allergy and Infectious Diseases (NIAID) Animal Care and Use Committee (ACUC) (protocol LMVR15). Animals were purchased from NIH-approved sources and transported and housed according to Guide for the Care and Use of Laboratory Animals [67].

### Animal studies

Four *Saimiri boliviensis* and 3 *Aotus nancymaae* New World monkeys were individually infected with 1 of 4 strains of *P. vivax* (Table 1 and S1 Fig): Chesson, a chloroquine sensitive parasite isolate from New Guinea [68]; Indonesia-I/CDC, a chloroquine resistant isolate from Indonesia [69,70]; AMRU-I, a chloroquine resistant isolate from Papua New Guinea [71]; or NIH-1993, a moderately chloroquine resistant strain closely related to the Salvadorian reference strain Salvador-I [72]. All monkeys were splenectomized before being infected intravenously with parasitized erythrocytes from cryopreserved stocks. Baseline hematocrit, weight, and blood smears were recorded from each animal before inoculation. After infection, hematocrit was measured once a week and parasitemia 3 times a week using approximately 0.05 mL of blood collected from the saphenous vein. Once parasites were observed on thin blood films, hematocrit was checked 3 times a week and parasitemia daily. Blood films were fixed with 100% methanol and stained for 15 minutes with a 20% Giemsa solution. For sample collection, blood draws from the femoral vein were performed on animals anesthetized with 10 mg/kg ketamine, using a laced heparin sodium syringe and a 25-gauge needle. The 3 *Aotus* monkeys received one dose of chloroquine 3 days after the initial blood sample collection, and we collected an additional blood sample 16 hours after this administration to assess the effect of chloroquine on *P. vivax* gene expression profiles (S1 Fig).

### Sample processing, library preparation, and sequencing

A total of 1 mL of fresh venous blood was obtained from each monkey and centrifuged at 2,500 rpm for 3 minutes to separate and remove the serum. Blood cells were transferred to 50 mL tubes containing RPMI 1640 to a final 0.5% to 1.0% hematocrit solution. Parasitized cells were enriched using MACS LS separation columns [35]. We determined the number and viability of the purified cells from each sample using a BioRad TC20 Automated Cell Counter and loaded, from each sample, an estimated approximately 3,000 cells on a 10X Genomics Chromium controller to prepare a barcoded 3'-end scRNA-seq library according to the manufacturer's instructions. For each of the 10 libraries, we minimized the amount of processing time

so that less than 3 hours elapsed between the blood collection and the cell isolation on the Chromium controller.

The 10 scRNA-seq libraries were sequenced with an Illumina HiSeq 4000 to generate a total of 1,254,873,695 paired-end reads of 75 bp (S1 Table). (One library was sequenced using 101 bp paired-end reads but was trimmed, before alignment, to 75 bp for consistency.) All sequence reads generated are available on NCBI SRA under the Bioproject PRJNA603327.

## scRNA-seq analysis

To process the scRNA-seq reads, we developed custom analyses mirroring the 10X CellRanger pipeline, allowing us to more easily control the different analysis parameters and to address the specificities of the *P. vivax* genome (see below).

We kept only reads containing the accepted 10X Genomics barcodes, and we trimmed scRNA-seq reads extending beyond polyadenylation tails by removing 3' sequences downstream of 6 consecutive As. All sequences shortened to less than 40 nucleotide long were discarded from further analyses (S1 Table). We then mapped all 1,091,808,431 remaining reads to the *P. vivax* P01 genome sequence [36] using HISAT2 (version 2.0.4 [73]) with the default parameters, except for a maximum intron length of 5,000 bp. We also separately mapped all reads to the *Saimiri boliviensis* (SaiBol1.0) or *Aotus nancymaae* (Anan2.0) genome sequences using HISAT2 and a maximum intron length of 20,000 bp.

To distinguish molecules amplified during the library preparation from genuine multiple copies of mRNAs present in one parasite, we determined for each sample which reads carried the same 16-mer 10X barcode, 10-mer randomizer sequence (or unique molecular identifier), and mapped within 10 bp of the same genomic location and on the same DNA strand. We randomly kept one occurrence of each unique molecule and discarded the others as likely PCR duplicates (S1 Table).

For each sample, we separated the reads derived from each individual parasite using the 10X barcodes. We then calculated the number of reads mapped to each nonoverlapping 500 bp window of each strand of the *P. vivax* P01 genome and annotated each window with regard to its position relative to the closest gene 3'-end on the same strand (based on PlasmoDB-37 annotations), using the following priorities to account for the gene-rich nature of the *P. vivax* genome and possible overlaps (S5 Fig): (1) containing an annotated gene 3'-end ("0"); (2) immediately following a window containing a gene 3'-end ("1"); (3) immediately preceding a gene 3'-end ("-1"); (4) 2 windows downstream of a gene 3'-end ("2"); (5) 2 windows upstream of a gene 3'-end ("-2"); (6) 3, 4, or 5 windows downstream of a gene 3'-end ("3," "4," and "5," respectively); and (7) further away from any annotated gene 3'-end ("X"). Finally, we discarded all individual parasite transcriptomes containing less than 5,000 unique reads mapped to the *P. vivax* genome as well as those containing more than 75,000 unique reads (S1 Table) and normalized the read counts across all parasite transcriptomes by dividing the read counts obtained for each window by the total number of mapped reads obtained for the given parasite. For the gene-based analyses, we combined reads from all windows associated with a given annotated protein-coding gene (with the caveat that this approach might collapse signals from different transcripts). To validate that the scRNA-seq reads mapping far from the 3'-end of annotated genes were genuine, we compared their mapping locations with the positions of the 3'-ends of transcripts predicted from the bulk RNA-seq data of a *P. vivax*–infected Cambodian patient (C0) using StringTie version 2.1.1 [74].

## PCA

To examine the diversity of, and relationships among, individual *P. vivax* parasite transcriptomes from the different infections, we conducted PCA using data from 13,054 windows containing at least 0.25% of the reads of one parasite, scaling the variance across windows. We then used the first 3 principal components to determine the position of each asexual parasite and female gametocyte along their putative developmental trajectories (later referred to as "pseudotime") by calculating their Euclidean distance from an arbitrarily selected origin (Fig 1A). Because the male gametocytes formed an apparently homogenous population without much variation along the PCs, we treated all male gametocytes identically in our analyses without calculating developmental pseudotime.

## Analysis of gametocyte genes

We calculated, for each individual parasite, the proportion of scRNA-seq reads originating from genes previously defined as characteristic of gametocyte rings (GRs), immature gametocytes (IGs), and mature gametocytes (MGs) [19].

We also screened the scRNA-seq data for genes expressed specifically in gametocytes by selecting all genes for which the mean expression in either male or female gametocytes was at least 10-fold greater than the mean expression in asexual parasites and in the other sex. We then verified that the selected genes were expressed in most gametocytes of the selected sex and absent from other stages. We used these gametocyte-specific genes to rerun the gene co-expression analysis on data previously generated by whole-blood RNA-seq from 26 Cambodian patients [20]: we calculated the Pearson's R correlation coefficient between the expression level of each pair of selected gametocyte genes and grouped these genes according to their co-expression among patients using unsupervised hierarchical clustering.

## Comparison with *P. berghei* scRNA-seq data

We compared the scRNA-seq data generated from *P. vivax* parasites to the data generated using 10X scRNA-seq for 4,884 *P. berghei* from in vivo bloodstream infections [34]. We used PlasmoDB (www.plasmodb.org) to identify 4,265 *P. berghei* genes with one-to-one ortholog to *P. vivax* and calculated for each parasite of each species the proportion of the mRNA molecules derived from this subset of genes. We then performed PCA jointly for the parasites of both species using all genes accounting for more than 1% of the mRNAs in at least one parasite and scaling the variance across genes. In addition, we recalculated the pseudotime for the *P. berghei* parasites as described above and using the principal component coordinates originally calculated by Howick and colleagues using all *P. berghei* genes and retrieved the stage annotations (i.e., rings, trophozoites, schizonts, and gametocytes) previously determined for each parasite [34].

## *P. vivax* genes with orthologs in the *P. falciparum* genome

We used PlasmoDB to identify the subset of *P. vivax* genes with at least one annotated orthologous gene in the *P. falciparum* 3D7 genome. We then calculated, for each individual *P. vivax* parasite, the proportion of scRNA-seq reads originating from genes with *P. falciparum* orthologs.

## Statistical analysis

To test for differential gene expression, we first normalized our data with *scran* [75] and calculated a size factor for each gene after pooling counts across cells to account for the low and

zero counts typical of scRNA-seq data. We then imported the count tables and the size factors to perform the statistical analyses with zinbwave [76] and corrected all statistical tests for multiple testing correction using FDRs [77,78].

To evaluate the effect of chloroquine on *P. vivax* gene expression, we compared the profiles of individual parasites of the same strain in the same *Aotus* monkey before and 16 hours after chloroquine administration. We conducted these analyses independently for each strain and separately for female gametocytes and asexual parasites, which were further subdivided into 4 crude developmental groups along the intraerythrocytic cycle. In addition, we used the pseudotime determined for each parasite as a covariate in our analyses to further correct for subtle developmental differences. Too few male gametocytes were detected in each sample to support rigorous statistical testing, and these were therefore not included in this analysis. We only considered as differentially expressed windows that were statistically significant (FDR < 0.1) for the same developmental group in all 3 strains and showed expression change in the same direction (e.g., they were all up-regulated upon chloroquine exposure).

We performed a similar analysis to evaluate the effect of the host species on the parasite gene expression and determine which *P. vivax* transcripts were differentially regulated in *Aotus* and *Saimiri* in each developmental stage. For this analysis, the different strains used in both monkey species and the low number of parasites obtained from several *Saimiri* infections prevented rigorous paired analyses (i.e., comparisons of expression differences between parasites of the same strain in the 2 hosts). We therefore pooled data from parasites obtained from all infections in a given host species together for this analysis.

## Genetic analysis

We identified single-nucleotide variants and sequence rearrangements from each *P. vivax* strain using the sequence information contained in the scRNA-seq reads. First, we examined nucleotide variation at each position of the *P. vivax* genome sequenced by at least 20 reads in a given sample. For each sample, we calculated the proportion of reads carrying the reference allele at each genomic position and examined the distribution of reference allele frequencies to assess the clonality of each sample [79]. We also investigated 2,279,246 nucleotides sequenced at >20X in all 10 samples to identify variable positions between parasites and calculated the number of nucleotide differences between the *P. vivax* parasites present in each pair of infections. Second, we calculated the number of reads mapped to nonoverlapping 10-kb windows for each sample and further examined windows without any read in one or more sample to detect possible chromosomal deletions (see S1 Text for details).

## Supporting information

**S1 Fig. Development of *P. vivax* parasitemia in nonhuman primates.** The graphs show the changes in parasitemia over time in *Aotus* (A) and *Saimiri* (B) monkeys infected with different strains of *P. vivax*. The large black circles represent, for each animal, the collection points of blood samples used for scRNA-seq library preparation. "CQ" indicates point of single oral administration of chloroquine to the animals (10 mg/kg for the Chesson and AMRU-I infections, 5 mg/kg for the Indonesia-I/CDC infection). Underlying data plotted in panels A and B are provided in S6 Data. scRNA-seq, single-cell RNA sequencing.
(TIFF)

**S2 Fig. Host and parasite contribution to the single-cell transcriptomes.** The scatter plot shows the number of monkey (x-axis, in 100,000s) and *P. vivax* (y-axis, in 100,000s) unique sequences (or UMIs) obtained from each droplet (gray dots) across all samples. Note that most

single-cell transcriptomes contain either monkey or *P. vivax* reads. The few droplets containing high numbers of monkey and *P. vivax* reads (dots off the x- and y-axes) are likely droplets containing multiple cells. UMI, unique molecular identifiers.
(TIF)

**S3 Fig. PCA of the 13,503 individual parasite transcriptomes containing more than 1,000 unique reads.** Each gray dot represents a single *P. vivax* transcriptome and is displayed according to its gene expression profile along the first 3 principal components. Note that the figure is virtually identical to the PCA generated using the 9,215 transcriptomes with more than 5,000 unique reads (Fig 1A). Underlying data are provided in S7 Data. PCA, principal component analysis.
(TIF)

**S4 Fig. Influence of the minimum read cutoff on the parasite stages retrieved.** The graph shows the number of parasites obtained (y-axis) using a minimum read cutoff of 1,000 (red) or 5,000 (green) unique reads per parasite. The parasites are grouped according to their pseudo-times (x-axis) calculated using the full data set and binned into 10 categories for the asexual parasites and female gametocytes from 0 (low pseudotime value) to 9 (high pseudotime value). Underlying data are provided in S8 Data.
(TIF)

**S5 Fig. Annotation of the 500-bp windows with regard to annotated *P. vivax* genes.** Each 500 bp window was annotated based on its distance from the 3'-end of the closest annotated gene. "0" corresponds to a window containing an annotated 3'-end, "1" through "5" indicate windows located 1 to 5 windows of 500 bp downstream of the annotated 3'-end, and "-2"/"-1" upstream of the 3'-end. "X" represents bins located outside of these regions. The annotation was performed independently for each DNA strand. The histogram on top shows the distribution of the windows containing scRNA-seq reads with regard to annotated genes. Underlying data are provided in S9 Data. bp, base pairs; scRNA-seq, single-cell RNA sequencing.
(TIF)

**S6 Fig. Example of scRNA-seq data generated using the 10X 3'-end protocol.** The figure shows approximately 20 kb on chromosome 11 between position 736,000 and 756,000. The blue bars at the bottom represent 7 annotated *P. vivax* genes in this region. The gray histograms represent the number of reads covering a given position for 24 single-cell transcriptomes generated from an infection of an *Aotus* monkey with AMRU-I. The first 16 rows correspond to asexual parasites along their developmental trajectories (by groups of 4), the next 4 rows represent female gametocytes, and the last 4 rows, male gametocytes. scRNA-seq, single-cell RNA sequencing.
(TIF)

**S7 Fig. Animated.mp4 showing the PCA of the 9,215 individual *P. vivax* parasite transcriptomes.** Each dot represents a single parasite transcriptome and is displayed according to its gene expression profile along the first 3 principal components. PCA, principal component analysis.
(MP4)

**S8 Fig. Animated.mp4 showing the PCA of the 9,215 *P. vivax* (in blue) and 4,884 *P. berghei* parasites (in pink).** Each dot represents a single parasite transcriptome and is displayed according to its gene expression profile along the first 3 principal components. PCA, principal component analysis.
(MP4)

**S9 Fig. PCA of the individual parasite transcriptomes separated by samples.** The figure shows PCA from all individual transcriptomes separated by blood samples. The rows correspond to the different strains of *P. vivax* used in this study, the columns blood samples collected from *Aotus* monkeys before and after chloroquine treatment (columns 1 and 2, respectively) or from *Saimiri* monkeys (column 3). Note that 2 different *Saimiri* monkeys were infected with the same strain of NIH-1993 (last 2 rows). Underlying data are provided in S10 Data. PCA, principal component analysis.
(TIF)

**S10 Fig. Maturity of *P. vivax* gametocytes circulating in blood.** The panels show the average proportion of transcripts (y-axis, in mRNA molecules per 1,000) derived from *P. vivax* genes associated with GR (red), IG (green), and MG (blue) based on Obaldia and colleagues, 2018. The left and middle panels show the expression in asexual parasites and male gametocytes, respectively. The right panel displays the average expression of 50 female gametocytes organized according to their pseudotime (left, differentiating; right, fully differentiated). Underlying data are provided in S11 Data. GR, gametocyte ring; IG, immature gametocyte; MG, mature gametocyte.
(TIF)

**S11 Fig. Correlations between the expression of male and female gametocyte genes.** The heatmap shows the extent of gene expression correlation (Pearson's R, in blue-red scale) between 2 gametocyte genes selected from scRNA-seq data across all Cambodian patients characterized in Kim and colleagues, 2019. The bordering trees show the result of unsupervised clustering of these genes according to their gene expression patterns. Underlying data are provided in S12 Data. scRNA-seq, single-cell RNA sequencing.
(TIF)

**S12 Fig. Expression of genes used for RDT.** The figures show PCA from all *P. vivax* transcriptomes colored according to the expression level of, from top to bottom, 2 genes used in RDTs, lactate dehydrogenase (PVP01_1229700) and fructose 1,6-bisphosphate aldolase (PVP01_1262200), and one gene almost ubiquitously expressed (profilin or PVP01_0730900). Each dot represents one single-cell parasite and is colored according to the expression level of that transcript (in red scale). Underlying data are provided in S13 Data. PCA, principal component analysis; RDT, rapid diagnostic test.
(TIF)

**S13 Fig. Expression of 28 AP2 domain transcription factors.** Variations in the expression levels of AP2 genes according to the parasite development. Each row represents a different annotated AP2 gene and each vertical bar an individual *P. vivax* parasite ranked along the x-axis based on its pseudotime. The color of each vertical bar represents the expression level of a specific transcript from nondetected (yellow) to highest expression (in red) in yellow-to-red scale. The color bar under the heatmap shows the assignment of the parasites into different morphological stages (red, trophozoites; blue and purple, schizonts; green, female gametocytes; orange, male gametocytes). Underlying data are provided in S14 Data.
(TIF)

**S14 Fig. Example showing possible alternative 3'-UTR usage.** The figure shows scRNA-seq data from 24 individual parasites for a region of approximately 10 kb on chromosome 9 containing 2 annotated *P. vivax* genes, PVP01_0941100 (left blue bar) and PVP01_0941200 (right blue bar), both encoded on the + strand. The scRNA-seq sequences generated from for asexual parasites (rows 1–16) map close to the annotated 3'-end of PVP01_0941100, whereas those generated from male gametocytes (last 4 rows) map more than 1 kb downstream. Female

gametocytes (rows 17–20) displayed an intermediate profile. This pattern is consistent with the presence of 2 different 3'-UTRs for PVP01_0941100 (although it is also possible that the second peak corresponds to the 3'-UTR of an unannotated *P. vivax* gene). scRNA-seq, single-cell RNA sequencing; UTR, untranslated region.
(TIF)

**S15 Fig. Expression of MSP7-like genes.** The figure shows examples of scRNA-seq data across approximately 30 kb of chromosome 12 containing a cluster of MSP7-like genes (blue bars at the bottom). Note that most MSP genes are specifically expressed in late asexual parasites (rows 5–12) but turned off in very late schizonts (rows 13–16), whereas other MSP7-like genes are uniquely expressed in female and male gametocytes (row 17–24). MSP, merozoite surface protein.
(TIF)

**S16 Fig. Expression of PHIST genes.** Variations in the expression levels of members of the PHIST multigene family according to the parasite development. Each row represents a different annotated PHIST gene, and each vertical bar an individual *P. vivax* parasite ranked along the x-axis based on its pseudotime. The color of each vertical bar represents the expression level of a specific transcript from nondetected (yellow) to highest expression (in red) in yellow-to-red scale. The color bar under the heatmap shows the assignment of the parasites into different morphological stages (red, trophozoites; blue and purple, schizonts; green, female gametocytes; orange, male gametocytes). Underlying data are provided in S15 Data. PHIST, plasmodium helical interspersed subtelomeric.
(TIF)

**S17 Fig. Heatmap showing the variation in the time of PIR gene expression.** Variations in the expression levels of PIR genes according to the parasite development. Each row represents a different annotated PIR gene and each vertical bar an individual *P. vivax* parasite ranked along the x-axis based on its pseudotime. The color of each vertical bar represents the expression level of a specific transcript from nondetected (yellow) to highest expression (in red) in yellow-to-red scale. The color bar under the heatmap shows the assignment of the parasites into different morphological stages (red, trophozoites; blue and purple, schizonts; green, female gametocytes; orange, male gametocytes). Underlying data are provided in S16 Data.
(TIF)

**S18 Fig. Heatmap showing the variation in the time of tryptophan-rich antigen gene expression.** Variations in the expression levels of tryptophan-rich antigen genes according to the parasite development. Each row represents a different annotated tryptophan-rich antigen gene and each vertical bar an individual *P. vivax* parasite ranked along the x-axis based on its pseudotime. The color of each vertical bar represents the expression level of a specific transcript from nondetected (yellow) to highest expression (in red) in yellow-to-red scale. The color bar under the heatmap shows the assignment of the parasites into different morphological stages (red, trophozoites; blue and purple, schizonts; green, female gametocytes; orange, male gametocytes). Underlying data are provided in S17 Data.
(TIF)

**S19 Fig. Proportion of mRNA transcribed from *P. vivax* genes with a *P. berghei* ortholog.** The figure shows the proportion of the mRNA molecules derived from *P. vivax* genes with a one-to-one *P. berghei* ortholog (y-axis) according to the parasite developmental stage (x-axis). Purple dots indicate asexual parasites organized according to their developmental pseudotime rank (from trophozoites on the left to schizonts on the right); green dots represent female

gametocytes and orange dots male gametocytes. Underlying data are provided in S18 Data.
(TIF)

**S20 Fig. Effect of chloroquine and host species on *P. vivax* gene expression.** (A) Expression of histone H4 (PVP01_0905800) in individual parasites organized along their developmental pseudotime rank (on the x-axis) and colored according to the chloroquine treatment (red, before treatment; blue, after treatment). Only parasite transcriptomes generated from *Aotus* infections are represented. (B) Expression of the exported protein PVP01_1402200 in individual parasites organized along their developmental pseudotime rank and colored according to the host species (red, *Aotus*; blue, *Saimiri*). The y-axis indicates the number of unique reads mapped to these windows in each parasite (in reads per 1,000). Underlying data are provided in S19 Data.
(TIF)

**S21 Fig. Possible deletion of DBP2 in monkey-adapted *P. vivax* strains.** The figure shows all the reads from each *P. vivax* strain mapped to a 50-kb region surrounding DBP2 (PVP01_0102300, highlighted in red). Virtually no reads mapped in the first 110 kb of this chromosomal arm (including DBP2) possibly indicating a deletion of this region in these monkey-adapted strains. Note that the boundary of this deletion might differ between strains (as shown by the location where read mapping resumed).
(TIF)

**S22 Fig. Reference allele frequency plots.** The figure shows reference allele frequency plots generated from each blood sample. Each plot shows the proportion of reads carrying the reference allele (x-axis, for 0 on the left to 1 on the right) for each position sequenced at >20X (y-axis). The rows correspond to the different strains of *P. vivax* used in this study, the columns blood samples collected from *Aotus* monkeys before and after chloroquine treatment (columns 1 and 2, respectively) or from *Saimiri* monkeys (column 3). Note that 2 different *Saimiri* monkeys were infected with the same strain of NIH-1993 (last 2 rows). Underlying data are provided in S20 Data.
(TIF)

**S1 Table. Summary statistics for the alignment and bioinformatics analyses.** The table shows, for each sample, the viability determined after MACS column purification, the number of reads generated, the percentage of reads carrying a 10X barcode (% Accept Barc), the percentage of reads too short after trimming polyA sequences (% Discarded), the percentage of reads mapped to the monkey and *P. vivax* genomes, the average number of reads per unique sequences (reads/UMI) and the number of single-cell transcriptomes obtained. UMI, unique molecular identifiers.
(XLSX)

**S2 Table. Comparison of the results from different *Plasmodium* scRNA-seq experiments.** For each study, the table indicates the number of single-cell transcriptomes obtained, the average number of unique reads mapped to annotated genes per parasite, the average number of expressed genes detected and the average number of genes contributing to more than 1% of the mRNA molecules. Data from Howick and colleagues, 2019, only include blood parasites characterized on the 10X Genomics platform. scRNA-seq, single-cell RNA sequencing.
(XLSX)

**S3 Table. Genes highly expressed in male and female *P. vivax* gametocytes.** The table shows the 25 genes most highly expressed in male and female gametocytes and provides for each genes the median and range (10%–90%) expression in female, male, and asexual parasites (in number of mRNA molecules per 1,000). The figures for *P. berghei* are also provided, as well as

whether the expression of the gene was detected in Yeoh and colleagues, 2017 (M in male, F in female gametocytes) and in Obaldia and colleagues, 2018 (GR, gametocyte ring; MG, mature gametocytes). Genes highlighted in green are specifically expressed in gametocytes.
(XLSX)

**S4 Table. Genes differentially expressed after chloroquine treatment.** The table lists the 500-bp windows deemed differentially expressed after chloroquine exposure (see main text for details). Cells highlighted in green represent genes up-regulated after chloroquine exposure, cells highlighted in red represent genes down-regulated after chloroquine exposure. The capital letters after the strain name indicate the stage of the parasite tested (A to D: asexual parasites from early trophozoites to late schizonts, F: female gametocytes). bp, base pairs.
(XLSX)

**S5 Table. Genes differentially expressed genes according to the host species.** The table lists the ten 500-bp windows most differentially expressed according to the host species at each stage (see main text for details). Cells highlighted in green represent genes up-regulated after chloroquine exposure, cells highlighted in red represent genes down-regulated after chloroquine exposure. The capital letters indicate the stage of the parasite tested (A to D: asexual parasites from early trophozoites to late schizonts, F: female gametocytes). bp, base pairs.
(XLSX)

**S6 Table. Genetic differences between strains.** The table shows the number of nucleotide differences between the different *P. vivax* strains used in this study. The cells highlighted in green indicate comparisons between the genotype from the same strain, either in the same monkey (before and after treatment) or in 2 different monkey infections.
(XLSX)

**S1 Text. Genetic Analyses.** Description of the assessment of clonality and the number of genetic differences between strains.
(DOCX)

**S1 Data. Experimental data used to generate Fig 1.**
(XLSX)

**S2 Data. Experimental data used to generate Fig 2.**
(XLSX)

**S3 Data. Experimental data used to generate Fig 3.**
(XLSX)

**S4 Data. Experimental data used to generate Fig 4.**
(XLSX)

**S5 Data. Experimental data used to generate Fig 5.**
(XLSX)

**S6 Data. Experimental data used to generate S1 Fig.**
(XLSX)

**S7 Data. Experimental data used to generate S3 Fig.**
(XLSX)

**S8 Data. Experimental data used to generate S4 Fig.**
(XLSX)

**S9 Data. Experimental data used to generate S5 Fig.**
(XLSX)

**S10 Data. Experimental data used to generate S9 Fig.**
(XLSX)

**S11 Data. Experimental data used to generate S10 Fig.**
(XLSX)

**S12 Data. Experimental data used to generate S11 Fig.**
(XLSX)

**S13 Data. Experimental data used to generate S12 Fig.**
(XLSX)

**S14 Data. Experimental data used to generate S13 Fig.**
(XLSX)

**S15 Data. Experimental data used to generate S16 Fig.**
(XLSX)

**S16 Data. Experimental data used to generate S17 Fig.**
(XLSX)

**S17 Data. Experimental data used to generate S18 Fig.**
(XLSX)

**S18 Data. Experimental data used to generate S19 Fig.**
(XLSX)

**S19 Data. Experimental data used to generate S20 Fig.**
(XLSX)

**S20 Data. Experimental data used to generate S22 Fig.**
(XLSX)

## Acknowledgments

We thank the technicians and veterinaries of the Division of Veterinary Resources, National Institutes of Health, for providing support to this study; S. Ott, Y. Santana-Cruz, L. Sadzewicz, and L. Tallon in the Genomic Resource Center at the University of Maryland School of Medicine for their support with 10X library preparation and Illumina sequencing; and A. Kim, R. Moraes Barros, and T. Gibson for helpful discussions and for critical comments on this manuscript.

## Author Contributions

**Conceptualization:** Juliana M. Sà, Thomas E. Wellems, David Serre.

**Formal analysis:** Juliana M. Sà, Matthew V. Cannon, David Serre.

**Funding acquisition:** Thomas E. Wellems, David Serre.

**Investigation:** Juliana M. Sà, Matthew V. Cannon, Ramoncito L. Caleon, David Serre.

**Methodology:** Juliana M. Sà, Matthew V. Cannon, David Serre.

**Project administration:** David Serre.

**Supervision:** Thomas E. Wellems, David Serre.

**Visualization:** Matthew V. Cannon.

**Writing – original draft:** David Serre.

**Writing – review & editing:** Juliana M. Sà, Matthew V. Cannon, Ramoncito L. Caleon, Thomas E. Wellems, David Serre.

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
