## [Editor Report · Decision Letter 0]

5 Jul 2019

Dear Dr Serre, 

Thank you for submitting your manuscript entitled "Single-cell transcriptomes of Plasmodium vivax parasites reveal extensive but continuous changes in gene expression during their intraerythrocytic cycle" for consideration as a Research Article by PLOS Biology.

Your manuscript has now been evaluated by the PLOS Biology editorial staff as well as by an academic editor with relevant expertise and I am writing to let you know that we would like to send your submission out for external peer review as a Resources paper if you agree.

**Important**: Please also see below for further information regarding completing the MDAR reporting checklist. The checklist can be accessed here: https://plos.io/MDARChecklist

Please re-submit your manuscript and the checklist, within two working days, i.e. by Jul 07 2019 11:59PM.

Kind regards,

Di Jiang, PhD

Associate Editor

PLOS Biology

INFORMATION REGARDING THE REPORTING CHECKLIST:

PLOS Biology is pleased to support the "minimum reporting standards in the life sciences" initiative (https://osf.io/preprints/metaarxiv/9sm4x/). This effort brings together a number of leading journals and reproducibility experts to develop minimum expectations for reporting information about Materials (including data and code), Design, Analysis and Reporting (MDAR) in published papers. We believe broad alignment on these standards will be to the benefit of authors, reviewers, journals and the wider research community and will help drive better practise in publishing reproducible research. 

We are therefore participating in a community pilot involving a small number of life science journals to test the MDAR checklist. The checklist is intended to help authors, reviewers and editors adopt and implement the minimum reporting framework. 

IMPORTANT: We have chosen your manuscript to participate in this trial. The relevant documents can be located here:

MDAR reporting checklist (to be filled in by you): https://plos.io/MDARChecklist

**We strongly encourage you to complete the MDAR reporting checklist and return it to us with your full submission, as described above. We would also be very grateful if you could complete this author survey:

https://forms.gle/seEgCrDtM6GLKFGQA

Additional background information:

Interpreting the MDAR Framework: https://plos.io/MDARFramework

Please note that your completed checklist and survey will be shared with the minimum reporting standards working group. However, the working group will not be provided with access to the manuscript or any other confidential information including author identities, manuscript titles or abstracts. Feedback from this process will be used to consider next steps, which might include revisions to the content of the checklist. Data and materials from this initial trial will be publicly shared in September 2019. Data will only be provided in aggregate form and will not be parsed by individual article or by journal, so as to respect the confidentiality of responses. 

Please treat the checklist and elaboration as confidential as public release is planned for September 2019.

We would be grateful for any feedback you may have.

---

## [Decision Letter · Decision Letter 1]

20 Aug 2019

Dear Dr Serre,

Thank you very much for submitting your manuscript "Single-cell transcriptomes of Plasmodium vivax parasites reveal extensive but continuous changes in gene expression during their intraerythrocytic cycle" for consideration as a Research Article at PLOS Biology. Your manuscript has been evaluated by the PLOS Biology editors, an Academic Editor with relevant expertise, and by three independent reviewers.

In light of the reviews (including the review from the Academic Editor), we would welcome resubmission of a much-revised version that takes into account the reviewers' and Academic Editor's comments as a Methods and Resources article. Our Academic Editor provides the following guidance to you on how to address the points raised by the reviewers.

Regarding reviewer 1's comments: You will need to address points 1 and 2 re normalisation, which are important. You will also need to provide a more in-depth comparison with the P. falciparum scRNAseq in response to point 4.

Regarding reviewer 2's comments: Paragraphs 2 and 3 re analysis method with a focus on genes/regions/windows analysed are important and must be addressed. Paragraph 4 suggests to extend and deepen the analysis to entire genome and comparison with other published scRNAseq of other plasmodium species is important and must be addressed. Paragraphs 5 and 6 are valid technical criticisms. Paragraph 8 suggests to exploit the scRNAseq thoroughly by looking for patterns of co-expression of TFs etc. This is important as reviewer 1 suggests the same in point 4. Both reviewers imply this identification of co-regulated genes is an additional analysis that would add considerable value. As it is the principal value that would raise the paper above the level of published bulk RNAseq, we ask you to meet this request from the reviewers. All minor comments are valid and should be addressed.

Regarding reviewer 3's comments: This reviewer has a large number of comments, most of which are reasonable and you will need to address them. The request for a de novo transcriptome would provide a useful resource. scRNAseq could allow the correct assignation of splice sites and utrs to specific isoforms if the coverage was adequate. There are very many putative AP2 binding sites in the P. falciparum genome and inferring a functional role for an over-represented, predicted DNA cis regulatory motif really requires functional analysis. However it is a simple analysis to do and detection of a shared AP2 binding motif would add value to inferred groups of co-regulated transcripts.

We cannot make any decision about publication until we have seen the revised manuscript and your response to the reviewers' comments. Your revised manuscript is also likely to be sent for further evaluation by the reviewers.

Your revisions should address the specific points made by each reviewer. Please submit a file detailing your responses to the editorial requests and a point-by-point response to all of the reviewers' comments that indicates the changes you have made to the manuscript. In addition to a clean copy of the manuscript, please upload a 'track-changes' version of your manuscript that specifies the edits made. This should be uploaded as a "Related" file type. You should also cite any additional relevant literature that has been published since the original submission and mention any additional citations in your response. 

Before you revise your manuscript, please review the following PLOS policy and formatting requirements checklist PDF: http://journals.plos.org/plosbiology/s/file?id=9411/plos-biology-formatting-checklist.pdf. It is helpful if you format your revision according to our requirements - should your paper subsequently be accepted, this will save time at the acceptance stage.

Please note that as a condition of publication PLOS' data policy (http://journals.plos.org/plosbiology/s/data-availability) requires that you make available all data used to draw the conclusions arrived at in your manuscript. If you have not already done so, you must include any data used in your manuscript either in appropriate repositories, within the body of the manuscript, or as supporting information (N.B. this includes any numerical values that were used to generate graphs, histograms etc.). For an example see here: http://www.plosbiology.org/article/info%3Adoi%2F10.1371%2Fjournal.pbio.1001908#s5.

For manuscripts submitted on or after 1st July 2019, we require the original, uncropped and minimally adjusted images supporting all blot and gel results reported in an article's figures or Supporting Information files. We will require these files before a manuscript can be accepted so please prepare them now, if you have not already uploaded them. Please carefully read our guidelines for how to prepare and upload this data: https://journals.plos.org/plosbiology/s/figures#loc-blot-and-gel-reporting-requirements.

Upon resubmission, the editors will assess your revision and if the editors and Academic Editor feel that the revised manuscript remains appropriate for the journal, we will send the manuscript for re-review. We aim to consult the same Academic Editor and reviewers for revised manuscripts but may consult others if needed.

We expect to receive your revised manuscript within two months. Please email us (plosbiology@plos.org) to discuss this if you have any questions or concerns, or would like to request an extension. At this stage, your manuscript remains formally under active consideration at our journal; please notify us by email if you do not wish to submit a revision and instead wish to pursue publication elsewhere, so that we may end consideration of the manuscript at PLOS Biology.

When you are ready to submit a revised version of your manuscript, please go to https://www.editorialmanager.com/pbiology/ and log in as an Author. Click the link labelled 'Submissions Needing Revision' where you will find your submission record. 

Sincerely,

Di Jiang, PhD

Associate Editor

PLOS Biology

Reviewer remarks:

Reviewer #1: The current manuscript describes application of a single-cell sequencing approach to characterising stage specific transcriptomes for blood-stages of Plasmodium vivax taken from experimental infected primates. The work is very interesting and the potential identification of stage-specific markers, particularly for gametocyte stages, is of high value. 

Major comments:

1. I have some significant concerns about the normalization strategy used in the study. The authors have mapped the data using standard and appropriate tools and removed PCR duplicate reads, which is appropriate and considered the current gold-standand approach for scRNA-seq data. However, to normalize the data, they have discarded low and high coverage outliers and then normalized their read count data (binned by mapping window) based on the total reads generated for the cell. This approach is essentially the "transcripts per million" transformation used in bulk RNA-seq analyses. This is fine, but even for bulk data, it is necessary to use more sophisticated normalization methods (e.g., TMM or UQ normalization) prior to TPM calculation, which are more robust to large differences in sequencing depth and other potential biases in RNA-seq data. 

For scRNA-seq, the preferred approach is to include quantitative spike-in controls to support data normalization. In the absence of this, it is necessary to use normalization strategies specifically suited to scRNA-seq data. Removal of high and low coverage outlier cells would help with this, but it is not considered enough on its own. A recent study hosted on BioRxiv (https://www.biorxiv.org/content/early/2019/03/19/583013.full.pdf) systematically evaluates scRNA-seq analytical approaches and highlights the appropriate data normalization as the most critical step in this process. In my opinion, the authors need to use a more robust normalization strategy, such as SCnorm (https://www.ncbi.nlm.nih.gov/pmc/articles/PMC5473255/) before reappraising the other steps in their study. I realise this is inconvenient and time-consuming, but I don't think it is possible to confidently make major conclusions based on the current analytical approach.

2. Following from this, the authors describing using edgeR for paired analyses of their data. This is not described in detail, but I take this to mean for differential gene expression analyses. If so, the details of these analyses need to be provided (What significance threshold was used? Did the authors used a Fischer's test or a linear model (i.e., Limma) for differentational calling?). Further and more importantly, EdgeR is a robust package for RNAseq and works very well for bulk RNAseq data, but it was not designed for scRNA-seq data specifically. In the absence of control spike-ins, it is critical that the data be appropriately and robustly normalized before using EdgeR (as noted above). A detailed comparative analysis of methods for calling differentially transcribed genes in scRNA-seq data can be found here, for example, doi:10.1038/nmeth.4612 and highlights other important steps including determining dropout rates and imputing missing gene values. For a example of specific applications in Plasmodium, the authors could also look at https://www.sciencedirect.com/science/article/pii/S0014482718306438, which used an Upper-Quartile normalization approach coupled with an scRNA-seq aware differential expression caller (SCDE). Reid et al, eLife, 2018 provides another good example and includes a approach for imputing missing gene data, which is an important step (https://www.ncbi.nlm.nih.gov/pmc/articles/PMC5871331/).

3. Results - page 8 - "Although accumulation of specific P. vivax stages has been reported in host tissues such as bone marrow the patterns, extent, and mechanisms of sequestration differ substantially from those of P. falciparum, and in many P. vivax infections, young as well as mature asexual and sexual stages can be found simultaneously in the circulation 27,40. Consistent with these observations, principal component analysis of the individual transcriptomes revealed distinct populations of blood stage parasites." - I'm not sure about the connection between the first and second sentence here. I understand the idea that P. vivax differs from P. falciparum in tissue sequestration patterns, but I don't see the direct connection to the distinct transcriptomic populations in the blood-stage data. Specifically, how do the authors differentiate that from simply different stage-specific transcriptomes? Do these population clusters differ markedly from the clusters seen in currently published scRNA-seq datasets for P. falciparum.

4. I feel the authors could make much more thorough use of available scRNA-seq data for blood-stages of Plasmodium. I understand the authors point in the introduction that prior studies have looked at cultured RBCs rather than cells harvested from an infected host. This is an important point, but in my view, that is all the more reason to compare the data more thoroughly. At present, the only comparison between P. vivax and P. falciparum in the current analysis is restricted to quantifying the proportion of the P. vivax transcriptome associated with genes that have or don't have a P. falciparum orthology. Why not also look at the relative stage specificity of these orthologs in existing scRNA-seq data? Ultimately, it is of course the authors' choice as to what they choose to include in their study. This is not a trivial undertaking and it may be beyond the scope of what the authors' wish to do here. I also am aware that nothing frustrates an author more than a reviewer commenting on a study they think the authors could have done, rather than the one they did do. That said, I think such an analysis would add to the impact of the study and is worth considering. 

Overall comments: This study is interesting and provides very valuable data that will be of broad interest to the field. I think identification of signals and markers of sexual-stage differentiation in P. vivax are important and again, will be of broad significance. However, the paper as written has some limitations in my view that must be addressed. Most significantly in the analytical stages in relation to the data normalization and differential gene expression analyses. These approaches are largely acceptable for bulk RNAseq data (with some caveats as noted above) but are not directly transferrable to scRNA-seq data. In my view, this has to be addressed before any further consideration of the manuscript can be made. I think the authors can make more of the comparative analyses with existing scRNAseq data for blood-stages of other Plasmodium species, but this is my subjective opinion and if the authors' choose not to expand the scope of their study, I won't stand in the way.

Reviewer #2: Major comments:

In general, the data for this paper are highly valuable and impressive but the paper somewhat suffers from an underwhelming use of the data that should be rectified before publication. 

This paper presents a bespoke analysis of P. vivax single cell data and would benefit from demonstrating that their approach is robust by comparing it to some of the more commonly used tools, for example at the very least presenting the CellRanger results. If the annotation of the genome is an issue and leads to poor results from CellRanger, then they could consider extending the annotated genes by a set amount from each start and stop (e.g., include 500bp up and downstream for each gene). It is unclear why they do not use popular and useful tools, and instead go about creating their own methods of analysis.

The authors’ use of windows rather than genes should be tempered by providing the gene-based analysis first. It is somewhat concerning that the screenshots of IGV seem to show transcription primarily outside of gene boundaries. This may very well be UTRs and indeed, the patterns of transcription that are presented support the notion that windows approach works, but it would still be good to provide insight into differences in patterns or indeed loss of information when using only annotated gene boundaries. Some stats on genes with and without annotated UTRs would also be important, as if UTRs are annotated, then the windows approach becomes less robust. 

The authors state that they have observed “exquisite regulation of P. vivax transcription along the intraerythrocytic life cycle” but support this claim with a heatmap of a small subset of genes. A more global analysis of the transcriptional changes over developmental time in needed. Do you observe abrupt patterns of expression as seen in other single cell Plasmodium data sets (Reid et al 2018), and are the timing of expression patterns similar to other species (e.g. Poran et al 2017, Howick et al 2019)?

The authors state that they have controlled for development by including pseudotime as a covariate. It’s important that they illustrate how this correction affects the results, as it seems to not capture how genes would be differentially expressed in a pseudotime dependent manner. 

The authors conclude that because the putative male population of cells doesn’t express genes associated with mature gametocytes that the known marker genes for males are not accurate. That these are definitely mature male gametocytes is unconvincing, and a more in-depth comparison to other published expression data sets in P. falciparum and P. berghei in needed including this recent biorxiv paper: van Biljon R, et al bioRxiv. 2019. p. 633222. doi:10.1101/633222.

Throughout the paper the authors focus on individual genes or subsets of genes but don’t include any functional follow up or global analysis to support their conclusions. For example the authors look at expression of the MSPs and PIRs but discuss them in the context of all gene families. Is this pattern generalizable? Can the authors comment on function about the other gene families given their patterns of expression?

DE assessments are not really using advantages of having single cell data at all, and presumably could have been done using bulk approaches with various purification methods to gather different stages. Consider fixing this. Also consider incorporating more of the power that single cell approaches give to examine things like variable gene expression, transcription factor coexpression, etc. 

Minor comments

1. Many minor grammatical issues throughout.

2. The statement in the introduction “Unfortunately, the requirement for fresh and viable parasites has so far precluded extension of such studies to other Plasmodium species” should say instead something about “unculturable human species”. Technically, this is also inaccurate as P.malariae is present in the Howick et al bioRxiv paper.

3. The authors state “We determined the number and viability of the purified cells” is stated in the methods, but no information is given on the results of the viability assays. Given the cells were enriched with MACS, the viability should be reported as well as when the viability assessments were made, given that three hours from sample to loading is a very long time. Were samples on ice this entire time?

4. Not enough detail on loading for 10x and how this was accounted for given different parasitemias; not enough detail on why the runs varied so much in cells recovered. 

5. Was there any attempt to get rid of WBCs? Provide a basic analysis of the pure monkey cells to understand if these are RBCs or WBCs.

6. The number of lanes of HiSeq 4000 used should be reported. 

7. hisat2 is more typically referred to as HISAT2 

8. Fig S7 should have the treatments/host labeled on it so that one can quickly identify which experiments are chloroquine treated, etc. 

9. Fig S10 purports to show that sexual and asexual parasites might use different UTRs for the same gene. From looking at the figure, it is also equally possible that those sexual parasites are just expressing the next gene along. 

10. “Most P. vivax genes remain unannotated..”, would be perhaps better stated as “..remain incompletely annotated”. With >6000 annotated genes in the P. vivax genome, it seems unlikely that there remain >6000 yet to be annotated. 

11. Fig 4 also needs some kind of normalizer that shows the total number of genes identified along pseudotime rather than just the proportion that have a P. falciparum ortholog.

12. The authors state “To statistically determine whether specific transcripts were differentially regulated upon chloroquine treatment, we compared the gene expression profiles of the parasites from the same Aotus infection before and 20-hour post- treatment, adjusting for differences in developmental stages” needs further elaboration on how you adjust for differences in developmental stages. 

13. No comment is made on the drug resistance status of the strains used; do any of them carry mutations associated with drug resistance? 

14. When discussing the different host species, it would be beneficial to remind the reader an estimate of how many rounds of re-invasion (length of time post inoculation) the parasites have been through. 

15. When referring to reference 57, which seems to investigate only Saimiri monkeys, it would be useful to discuss the locations from where the various P. vivax strains under discussion originate from. Indeed, this information appears to be completely missing from the methods/materials and I can only find strain names in table 1, with no explanation on what these are or where they originate from. 

16. Figure S4 is meant to show that the data do not change based on the QC cut off on reads per cell. However, it only demonstrated that the general shape of the PCA is the same (and is really hard to see the points in grey). It would be helpful to quantify how each developmental stage or cluster is represented depending on the cut off used. 

17. Figure 3 illustrates non-overlapping expression of 4 AP2 transcription factors. How many of the ~27 AP2s were detected and how common was this stage-specific expression?

18. Figure 5: label axes 

19. The authors state “Finally, we discarded all individual parasite transcriptomes containing less than 5,000 unique reads mapped to the P. vivax genome as well as those containing more than 75,000 unique reads (Table S1)” & “Figure S4. Principal component analysis of the 13,503 individual parasite transcriptomes each containing more than 1,000 unique reads. Each dot represents a single parasite transcriptome and is displayed according to its gene expression profile along the first three principal components. Note that the figure is virtually identical to the PCA generated using the 9,215 transcriptomes with more than 5,000 unique reads (Figure 1).”....These cut offs seem high - which cells do you lose with this filtering? Is it cell type biased?

20. The authors state “The analyses described above did not reveal any major effect of chloroquine (or of the host species) on the overall patterns of gene expression: we observed all parasite developmental stages in most samples regardless of the treatment or the host, and the transcriptomes of treated parasites or parasites from different hosts overlapped apparently perfectly (Table 1 and Figure S13).” and from methods: “We conducted these paired analyses using EdgeR 36, independently for female gametocytes and asexual parasites, and further subdivided the latter into four crude developmental categories along the intraerythrocytic cycle”. It is important to go into more detail of how you divided asexuals, and a single-cell specific differential expression tool should be used to account for dropout.

Reviewer #3: Overview:

The authors should be commended for their heroic undertaking to determine several Plasmodium vivax transcriptomes from in vivo infections using single cell RNA-seq. These are the first example of such data which builds on data the group has previously generated by replacing bulk transcriptomics from patient isolates with single cell RNA-sequencing data generated from the infected reticulocytes of splenectomized monkeys. Using dimensional reduction based analysis of the data, the authors are able to characterize the probable sequence of gene expression events throughout P. vivax asexual development and to some extent during male and female gametocytogenesis. Despite the potential richness of the datasets acquired in this study, the manuscript fails to deeply analyze the data, providing a very broad over-view of the P. vivax in vivo transcriptome that ultimately fails to deliver on new biological insights, rather confirming previous studies. The manuscript also appears to have been put together hastily and therefore is difficult to read at times and repetitive in other sections, lacking significant editing prior to submission.

General Comments:

It is difficult to understand from the manuscript what new information was learned relative to The transcriptome of Plasmodium vivax reveals divergence and diversity of transcriptional regulation in malaria parasites (2008 PNAS). Is the novelty in the unique markers that cannot be picked up by bulk RNA-seq? Furthermore, since in the previous study the authors could culture the parasites ex-vivo and therefore determine the morphology of the parasite unambiguously, the authors should compare their data to this study to see if the highest correlations exist at the expected stage (i.e. the trophozoite and schizont datasets correlate highly, whereas the ring data is missing as they speculate is the case in their sc-RNA-seq asexual transcriptome?). Furthermore, how does the proportion of genes detected in this sc-RNA-seq study compare to bulk studies? At what threshold do you no longer detect genes that previous datasets have determined to be expressed but at a low abundance?

Can the authors graph their phaseogram against that of the orthologous genes in a better characterized Plasmodium spp. (I.e. falciparum or berghei) to highlight the differences and similarities across malaria parasites?

Can the authors use the high quality paired end reads they have generated to assemble a putative P. vivax transcriptome de novo? It would be both interesting and of great use to the field to have a more reliable P. vivax transcriptome to map NGS sequencing data to, or conversely to provide higher confidence in the currently predicted gene models.

There is a missed opportunity in the authors’ exploration of the expression of AP2 domain proteins throughout asexual and sexual development. 12/27 of the AP2 domain proteins predicted in P. vivax have been genetically characterized to some extent in other Plasmodium species. How do their temporal expression patterns match their genetic roles? We feel that the PfAP2-I orthologue may be of particular interest due to its role in activating transcription of genes needed for red blood cell invasion in Pf. Although the essential AP2 domain (D3) of PFAP2-I is 100% conserved between Pv and Pf (implying that it will bind a similar DNA motif) the Pv invasion pathway is presented as being different and presumably involves the transcriptional activation of different genes, as the authors have pointed out.

In keeping with the above content, can you use the existing Pv gene models to predict DNA cis-regulatory motifs based on the temporal profile of mRNA abundance? This would provide a very convincing argument that the precisely regulated timing of transcription factor expression is relevant for controlling the rest of the transcriptome, as the authors have suggested.

Specific Comments:

Page 2 line 18- In the following citation, ex vivo cultured P. falciparum isolates were used to address some of the shortcomings mentioned, specifically the fact that cultured parasites lose the ability to express genes that are important for host interaction. 

Blythe, J. E., Xue, Y. Y., Kuss, C., Bozdech, Z., Holder, A. A., Marsh, K., … Preiser, P. R. (2008). Plasmodium falciparum STEVOR proteins are highly expressed in patient isolates and located in the surface membranes of infected red blood cells and the apical tips of merozoites. Infection and Immunity

Page 2 line 17- Was there any way for the authors to look at their parasites microscopically before generating the material for an RNA-sequencing library (as in reference 29)? How robust is the pseudotime data reduction i.e. can the age and life stage of parasites analyzed in this manuscript be said to be known with absolute certainty?

Page 4 line 21- how does mapping the artificially trimmed 75bp vs 101 bp reads compare? If you use the longer reads do you increase your information about gene structures?

Page 5 sequence analysis- Is it normal with single cell transcriptome data to normalize read depth across bins as you have described? Is there a significant difference in the sequencing depth originating from different locations on the transcriptome? Can you provide a sc-RNA seq citation that analyzes the reads in a similar way?

Can the authors provide an updated putative P. vivax transcriptome based on the gene structures (splice junctions, UTR coverage, etc.) detected in their data? Although that the goal of this work was to quantify transcripts using the 3’ end reads, the dataset should also impart some new insights into the transcriptome?

Page 7- The Results section describing “detailed transcription information” contains numerous arbitrary facts concerning the number of total reads and fraction pertaining to host versus P. vivax. In the end, this serves to confuse the reader more than providing clarity and diminishing the significance of the number of reads attained in this study. This section also describes a rational for inclusion of certain transcriptomes and not others which comes across as arbitrary without any statistical metric for inclusion/exclusion.

Page 7 line 25-29- Isn’t the information provided here contradictory? How can the spread of reads mapped to Pv be 40 to 80% but the majority of datasets are either all host or all Plasmodium?

Page 8 line 16- Please state how many genes 19,282 unique sequences corresponds to in your dataset.

Page 9 line 10- Is there a citation for the group of male gametocyte genes in P. vivax? Otherwise how were these predicted?

Page 9 line 20- How deep of sequencing do you need to detect SNPs with high confidence? Does this data consistently reach that threshold?

Page 9 – How were the genes selected for inclusion in Figure 2?

Page 10 line 20- Is it possible that the sc-RNA seq method does not adequately detect low abundance transcripts, and this is why previously theorized markers were not detected? What fraction of predicted P. vivax transcripts were detected relative to the previous bulk RNA-seq study in reference 18?

Page 10 line 31- P. vivax has direct orthologs to ap2-g2 and ap2-g3, which are thought to be important for gametocyte development in all Plasmodium species where they have been genetically interrogated. Are their transcripts detected in either gametocyte population? It would be useful to provide a comprehensive list of all predicted gametocyte-related genes (many are now available in the literature) and list their P. vivax orthologues to better define new versus previously suspected sexual stage genes emerging from this study.

Page 11- I believe that AP2 domain proteins should be written as AP2 (or ApiAP2), not AP-2

Many of the genetically characterized ApiAP2 proteins in Pb and Pf have orthologues in Pv and the important invasion transcription factor AP2-I has 1:1 conservation in its 3rd AP2 domain, implying that it probably binds a similar DNA sequence. Can you find any temporal conservation of potential 5' cis regulatory DNA elements in your data that implicate any of the AP2 domain proteins in the observed expression cascade? This may be especially interesting for AP2s involved in invasion (such as AP2-I) because or the differences in invasion mechanism between Pf and Pv.

Bozdech et al. previously proposed a cascade of regulation through AP2 proteins in their in vitro P. vivax transcriptome paper (2008).

Page 12- The authors state that the current gene annotations are rather poor for the P. vivax genome? Is this actually correct? What is the current number of gaps in this genome and by homology and synteny alone, can’t the counter argument be made that the genome is in quite good shape?

Page 12-13- Can you please provide a biological rationale why there would be a dramatic variability in the number (up to 63.4%) of genes expressed that are absent in P. falciparum during invasion? I can appreciate that invasion would be different for Pv, but why should it be so different across the measured cells? Some don’t appear to vary much at all, while others vary a lot.

Page 13 line 10- what does “apparently perfectly” mean in this context? Is the correlation between the datasets really 100%? There is no statistical basis for this statement.

Page14- if the authors optimized their isolation to find reticulocytes with mature parasites could the undetected subpopulations (related to the Duffy Binding Protein) just not be enriched in the purification process, as the authors already suggested for rings?

Figure 1- please add the gene names to the Gene ID (e.g. AMA1, Pvs25, etc…).

Figure 4- please clarify why there are different scales for each of the figures?

Miscellaneous-Pv is known to react differently to chloroquine than falciparum. Is there any reason to expect a transcriptional response that can be explained by falciparum biology?

Academic Editor's remarks:

The work uses excellent cutting-edge technology to report several novel findings and several that confirm what was already known from bulk RNAseq or scRNAseq in other species

The expression of Pv specific genes primarily in late stage schizonts for invasion is an important and novel finding.

Interestingly the authors show that there appears to be a deletion in monkey passaged paras and that an msp3 is differentially expressed between monkey spp, this evidence of species differences could be informative for factors specific for human disease.

The lack of co-variation between female and male gam genes is interesting, ie the ratios are diff between patients. I find the observation sound although it is confirmatory of what the authors previously published in their bulk rnaseq data. However, I found the terminology used here confusing 

“the terminal phases of male and female gametocytogeneses are independently regulated in P. vivax” 

Actually they seem to be dependently regulated, so that the probability of one cell being female affects another cell in the same host hence the disparity in sex ratios between hosts, if they were truly independent wouldn’t they trend to 50/50 or have the same unequal ratio across all hosts? 

H4 expression increased in cq treated paras, this is interesting in light of the known DNA intercalating properties of Cq but a lot of work over a long time has convincingly shown that the Cq kills via interfering somehow with haemoglobin digestion/hemozoin formation in the digestive vacuole so the authors need fairly strong evidence to propose resistance via increased histone expression, I think this section should be toned down. 

Interestingly the alternative histone H3.3 is upregulated (Table S3), H4 might be upregulated because they are deposited as a dimer together post replicatively. H3.3 is involved in gene regulation in many species and in telomeric silencing but in P falciparum it has some dynamic association with promoters but is primarily constitutively associated with subtelomeric repeats and coding sequence (Fraschka 2016 SciReports). It is implicated in male gamete development in mammals so could be somehow associated with gam development, this hasn’t been investigated. Increased H3.3-H4 incorporation could be a stress response to Cq. 

Two substantial aspects of the work were the ap2 lifecycle associations and the gender specific genesets. However I did not get a feel from the MS how different the vivax data were to already published datasets from other species. I think this needs to be more clearly discussed in light of the author’s own discussion of the need for P vivax studies due to its unique biology.

The expression of specific ap2 at diff stages suggesting they specifically regulate progression through the lifecycle. However the bulk culture RNAseq of P. berghei, (which is more closely related to P vivax than is P falciparum, although still quite distant) has already revealed the phased expression of sets of Api AP2 transcription factors (Modrzynska et al 2016 Cell Host Microbe) and importantly 12 of 26 ApiAP2 have been KOd/KDd and their role in stage specific development confirmed by functional evidence of lethality in various stages in this and other studies. The principle was thus established without the scRNAseq although not at the same resolution. The authors should discuss their findings in light of these other studies, are there results congruent with the other studies in other species?

“We next searched the scRNA-seq data to identify transcripts highly expressed in male and/or female gametocytes and with little expression in other stages to generate a comprehensive list of putative biomarkers for male and female gametocytes (Table S2).”

Similar analyses have been conducted using scRNAseq in P. falciparum and P berghei (Reid et al 2018 elife). Also bulk RNAseq of male and female gams in P berghei (Yeoh BMC genomics 2017) and P. falciparum (lasonder 2016 NAR). It would be useful to know how much of these genesets intersected with the current study and how much is unique to P vivax, also the extent to which scRNAseq identified novel sex specific genes compared to the earlier bulk RNAseq experiments. The findings of these previous study should be discussed by the authors.

Minor comments

the PCA plot in fig 1 is color coded by genes, the function of which is used as a surrogate of lifecycle stage but the function of the genes and their lifecycle association is not explicitly stated in the fig, its legend or the text.

Page 10 

“Interestingly, male gametocytes did not display elevated expression of genes previously associated with mature gametocytes (Figure S8), suggesting that these genes might be expressed only in female mature gametocytes”

This is not apparent to me from fig s8, it looks like the proportion of mature gam genes is as high as immature gam genes in males and the ratio of immature to mature is higher in males than females. The actual levels of expression of the mature gam genes in males does not seem to be presented in fig s8 at all.

Page 14 dbp2 nomenclature should be clarified in the text, its called ebp2 in the fig legend and ebp in the fig

Table S3 colour coding re significance seems odd, the numbers are ranks for the genes significance of fold change but this is not informative to me, they may be highly ranked and still non-significant, the colour coding isn’t explained, I assume they meet some significance threshold?

---

## [Decision Letter · Decision Letter 2]

31 Jan 2020

Dear Dr Serre,

Thank you very much for submitting a revised version of your manuscript "Single-cell transcription analysis of Plasmodium vivax blood-stage parasites identifies stage- and species-specific profiles of expression" for consideration as a Methods and Resources at PLOS Biology. This revised version of your manuscript has been evaluated by the PLOS Biology editors, the Academic Editor and two of the three original reviewers who also assessed your response to the comments of original reviewer 3.

In light of the reviews (below), we are pleased to offer you the opportunity to address the remaining points from reviewer 2, and our Academic Editor's comments which are included as Academic Editor's remarks at the bottom of this letter, in a revised version that we anticipate should not take you very long. Our Academic Editor provides the following guidance to you on how to address the points raised by the reviewer 2, which is included here.

“…The authors should provide at least one quantitative comparison of results using their bespoke method and CellRanger with UTRs extended in the GTF by a reasonable amount. This can be supplementary, and their main text can stay as it is, but it is important to do this”

Our Academic Editor suggests that you have mounted a convincing case as to why you need to use your bespoke method. 

“The new Figure S2 appears to indicate that >40% of observed transcription is more than 2.5kb away from an annotated gene. Is this true? It would be important to further discuss this in the manuscript if so, and draw comparisons with results from bulk RNAseq. Additionally, for the remaining windows that are within 1kb of a gene, it is important to do a more clear-cut gene-for-gene comparison to the other species scRNAseq data out there, as noted in the next section’s comment .”

Our Academic Editor indicates that this concern needs to be addressed and agrees with the reviewer that the bulk RNAseq should reveal roughly the length of 3’ UTRs which would validate your windows approach.

“…It would be reassuring to see that the reads that map to these new 3’ UTR regions in Pv map to 3’ UTRs in orthologous genes”

Academic Editor suggests that if orthologous means the 3’ UTRs in scRNAseq in other plasmodium spp, this wouldn't be very informative as the 3’ UTRs may be of different lengths. Another interpretation is the reviewer meant paralogs within Pv; finally they might mean the same genes in other Pv transcriptomes; the last interpretation seems reasonable but seems to reiterate the comment in the previous paragraph.

“The comparison of the shape of the PCA is a very superficial level look at similarity in expression, it is helpful to have as a figure, but more needs to be done, especially given the new title stating the paper is about species-specific expression profiles.”

Academic Editor agrees that the PCA plot is suggestive of similarity of profile but is not informative of the number or identity of similarly regulated genes across the life cycle. The new discussion and your claims of a gradual progression through transcription are useful, but our Academic Editor indicates that some more detailed comparison with other species is warranted. 

“Concerns raised by reviewer 3 could be alleviated by correlating expression of the single-cell transcriptomes to those of ex vivo Pv bulk data. This would provide more insight into what the pseudotime values actually correspond to in real time. If the data does not correlate well with bulk, the authors have outlined a clear reason (lack of synchronicity or washed out by trophs) above that would be an interesting discussion point.”

Academic Editor isn't convinced that this comparison would be useful to address this issue. Pseudotime has been accepted widely as a measure and, as the reviewers acknowledge, a disparity between pseudotime here and bulk RNAseq elsewhere would not reveal whether the pseudotime or the bulk RNAseq were incorrect and thus would not be very useful.

“There are many other methods to cluster genes besides WGCNA that might provide more clarity into patterns of coexpression. Saelens et al 2018 is a good review of the many different options (https://www.ncbi.nlm.nih.gov/pubmed/29545622). This could provide clusters that would allow for the identification of enriched motifs. Additionally, if greater than 40% of transcription is not assigned to a gene, then assigning regulatory regions will be a difficult task, so further clarification in the manuscript about genes in particular is required. If the annotations are so poor still that genes cannot be assigned, then this needs to be far clearer throughout the manuscript and explored. It does not mean that the results are not sound, but the “windows” based approach is confusing and needs to be more clearly explained why it is necessary and where possible, demonstrate that it works well (e.g., when windows can be clearly assigned to genes, the genes make sense ).”

Academic Editor believes that this reiterates the reviewer's concerns re the validity of the windows approach and suggests that you address this by comparing your RNAseq to Pv bulk RNAseq to confirm the validity of the predicted 3’ UTR length.

We will assess your revised manuscript and your response to the reviewers' comments and we may consult the reviewers again. We expect to receive your revised manuscript within 1 month.

Sincerely,

Di Jiang

PLOS Biology

REVIEWS:

Reviewer #1: I thank the authors for their considered and thorough responses to my comments. In my view, they have addressed these in detail and I have no further concerns with the manuscript. Per the editors request, I have viewed the comments also from reviewer 3 (who I understand was not available to review the revised manuscript). In my opinion, the authors have also addressed these comments in detail and appropriately, such that I have no further concerns about these comments either. Based on this, I am happy to recommend the study be accepted for publication. I don't have any further comments that require attention and I wish to congratulate the authors on an excellent contribution to the field.

Reviewer #2: Please see attachment for easier reading as it has colors...pasted here again in case attachment does not work.

The authors have addressed the majority of comments well, but some important areas still require further clarification and possibly work. We have pasted the original reviewer comments and author responses that we think require further consideration or work below. Our additional comments are in red. 

Reviewer 2 

This paper presents a bespoke analysis of P. vivax single cell data and would benefit from demonstrating that their approach is robust by comparing it to some of the more commonly used tools, for example at the very least presenting the CellRanger results. If the annotation of the genome is an issue and leads to poor results from CellRanger, then they could consider extending the annotated genes by a set amount from each start and stop (e.g., include 500bp up and downstream for each gene). It is unclear why they do not use popular and useful tools, and instead go about creating their own methods of analysis.

We might have incorrectly conveyed the idea that our analysis pipeline was very different from CellRanger while it is heavily based on it and conceptually identical (we have now clarified this point on page 4). We initially spent a fair amount of time trying to get CellRanger to work on our data and would have preferred to use this approach directly. We eventually chose to write our own pipeline, recoding the different steps implemented in CellRanger to allow us to more easily modify and control every parameter. As the reviewer accurately notes, the gene annotation was one key issue driving this choice: as indicated in the manuscript, only 42-48% of the reads mapped within 500 bp of the annotated 3'-end of a gene. In addition, we wanted to be able to analyze the precise genomic location of the scRNA-seq signals and avoid collapsing signals that might represent different transcripts (as shown for example on Supplemental Figure S14). Arbitrarily extending the 3'-end of each gene, as suggested by the reviewer, does improve the number of reads assigned to genes but does not alleviate our other concerns, nor allows conducting all the analyses described in the manuscript. We were also concerned that the criteria used by CellRanger for discarding cells with too few reads (all cells with 1/10th of the reads of the 99tile cells with the most expression) might fail to capture the range of transcriptional activity observed along the intraerythrocytic cycle (this concern turned out to be valid as Howick et al. 2019 had to manually "rescue" some cell populations that were systematically discarded by the CellRanger's thresholding). 

The improved method sections provides clarity on why this bespoke analysis was performed; however, it would still increase the impact of the paper if there was a more in depth comparison between the CellRanger output and the window based count table. How much of the biology that is discussed in the manuscript is actually lost by using CellRanger? Should all Plasmodium RNA-seq studies on P. vivax use this new method from now on? The authors should provide at least one quantitative comparison of results using their bespoke method and CellRanger with UTRs extended in the GTF by a reasonable amount. This can be supplementary, and their main text can stay as it is, but it is important to do this. 

The authors' use of windows rather than genes should be tempered by providing the gene-based analysis first. It is somewhat concerning that the screenshots of IGV seem to show transcription primarily outside of gene boundaries. This may very well be UTRs and indeed, the patterns of transcription that are presented support the notion that windows approach works, but it would still be good to provide insight into differences in patterns or indeed loss of information when using only annotated gene boundaries. Some stats on genes with and without annotated UTRs would also be important, as if UTRs are annotated, then the windows approach becomes less robust.

Because of the design of the 10X assay, which captures mRNAs with oligo-dTs and ligates the sequencing adapters after cDNA fragmentation, almost all scRNA-seq reads derive from the ~300 bp immediately preceding the polyA tail and therefore map to the 3'-UTRs (and not to the core of the gene). Given the incomplete annotation of the 3'-UTRs in P. vivax (none of the P. vivax P01 genes have 3'-UTR longer than 10 bp), it is not surprising that most reads map outside genes. Limiting our analyses to the current gene annotations would not only dramatically reduce the number of useful reads but could also collapse signals from different (unannotated) transcripts together and muddle the analyses. We did however summarize the bin data by genes and present these findings to facilitate the interpretation of the data (with a cautionary note on page 5). 

The new Figure S2 appears to indicate that >40% of observed transcription is more than 2.5kb away from an annotated gene. Is this true? It would be important to further discuss this in the manuscript if so, and draw comparisons with results from bulk RNAseq. Additionally, for the remaining windows that are within 1kb of a gene, it is important to do a more clear-cut gene-for-gene comparison to the other species scRNAseq data out there, as noted in the next section's comment. 

A better way to display information currently contained within S2 would be a conventional coverage plot across all genes including multiple windows beyond the stop codon. If the authors think the unannotated transcripts that are more than 500 bp from an annotated gene are driving the biology in their data, more work needs to be done on when and where they are expressed. It would be reassuring to see that the reads that map to these new 3' UTR regions in Pv map to 3' UTRs in orthologous genes.

The authors state that they have observed "exquisite regulation of P. vivax transcription along the intraerythrocytic life cycle" but support this claim with a heatmap of a small subset of genes. A more global analysis of the transcriptional changes over developmental time in needed. Do you observe abrupt patterns of expression as seen in other single cell Plasmodium data sets (Reid et al 2018), and are the timing of expression patterns similar to other species (e.g. Poran et al 2017, Howick et al 2019)?

We thank the reviewer for bringing this point to our attention. Reid et al., 2018 describes three distinct clusters of genes based on their expression in asexual parasites and interpret this pattern as indicative of "abrupt changes" in transcription, contrasting with the continuous cascade of transcription and smooth transition observed by bulk-RNA seq (see e.g., Ho et al, 2016 or Bozdech et al, 2003). However, while their figure 4C seems to show distinctive clusters of genes, the PCA in Reid et al shows a continuous distribution of the asexual parasites and not discrete clusters as we would expect if there were "abrupt" changes in expression. The heatmap shown in their Figure 4 also seems to indicate that each gene is not either on or off but gradually increasing or decreasing in expression (though we would agree that this is quite subjective). The three clusters of genes described in Reid et al., 2018 are different from what we observed in our data (e.g., our new Figures 2A, 3 and 4) and we wonder if this could be caused by an uneven distribution of the parasites along the intraerythrocytic cycle in the Reid study which could introduce some apparent clustering. We have included a comparison between our data and the recent P. berghei data generated by Howick et al. with the same technology as used in our study. This analysis highlights the similarities in transcriptional changes between these species along the intraerythrocytic cycle (see e.g., Figure 1B) and, we think, the conserved continuous changes in gene expression throughout the parasites' development. We have also included a short discussion on this aspect on pages 13-14. 

Although the authors have included a comparison with the P. berghei data from Howick et al, as well as some technical comparisons with other single-cell plasmodium data sets, they do not address what is differentially expressed over development in a species-dependent manner, i.e. do the same genes show the same pattern of expression across both Pb Pf, and Pv? The comparison of the shape of the PCA is a very superficial level look at similarity in expression, it is helpful to have as a figure, but more needs to be done, especially given the new title stating the paper is about species-specific expression profiles. 

Reviewer 3 

Page 2 line 17- Was there any way for the authors to look at their parasites microscopically before generating the material for an RNA-sequencing library (as in reference 29)? How robust is the pseudotime data reduction i.e. can the age and life stage of parasites analyzed in this manuscript be said to be known with absolute certainty?

We prepared blood smears from each sample and counted an estimated 10,000 red blood cells prior to processing through MACS columns. We agree that it would have been better to also have a second smear after MACS enrichment, but we did not have enough sample from the column. We have now included in Table 1 the stage composition determined by blood smear from each sample (prior to MACS enrichment). 

Regarding the calculation of the pseudotime, we believe that this metric captures most of the variation occurring along the developmental cycle. The absolute value of this estimate is probably not very informative (as it depends on changes in gene expression and on the distribution of the parasites along the intraerythrocytic cycle) but the ranking reflects morphological changes in the parasites, appears robust with regards to the data cutoffs and normalization (see above), and is correlated with the gene expression variation observed in P. berghei. 

Concerns raised by reviewer 3 could be alleviated by correlating expression of the single-cell transcriptomes to those of ex vivo Pv bulk data. This would provide more insight into what the pseudotime values actually correspond to in real time. If the data does not correlate well with bulk, the authors have outlined a clear reason (lack of synchronicity or washed out by trophs) above that would be an interesting discussion point. 

In keeping with the above content, can you use the existing Pv gene models to predict DNA cis-regulatory motifs based on the temporal profile of mRNA abundance? This would provide a very convincing argument that the precisely regulated timing of transcription factor expression is relevant for controlling the rest of the transcriptome, as the authors have suggested.

We tried to identify clusters of co-expressed genes (using a modified version of WGCNA) to then search for enriched motifs in the 5' region of each gene but failed to obtain convincing modules (probably due to the high number of drop-offs). At this point, we feel that our current data are not sufficient to address this question. 

There are many other methods to cluster genes besides WGCNA that might provide more clarity into patterns of coexpression. Saelens et al 2018 is a good review of the many different options (https://www.ncbi.nlm.nih.gov/pubmed/29545622). This could provide clusters that would allow for the identification of enriched motifs. Additionally, if greater than 40% of transcription is not assigned to a gene, then assigning regulatory regions will be a difficult task, so further clarification in the manuscript about genes in particular is required. If the annotations are so poor still that genes cannot be assigned, then this needs to be far clearer throughout the manuscript and explored. It does not mean that the results are not sound, but the "windows" based approach is confusing and needs to be more clearly explained why it is necessary and where possible, demonstrate that it works well (e.g., when windows can be clearly assigned to genes, the genes make sense).

Academic Editor's remark on your response to their first remark:

No fig legends for figs s7 and s8 

The authors meaning behind the lack of co-variation between male and female gam gene expression is now clearer that they are not using words with mathematical meaning in a non-mathematical fashion but I’m not convinced that the data supports the conclusion. I agree that, for example, high levels of female gam gene expression can occur with low, high or medium levels of male gam gene expression but I don’t see why this would necessitate separate processes of gametocytogenesis. An alternative explanation consistent with the data is a common pathway of sexual differentiation that defaults to one sex, then those that fail to switch to the other sex will end up the default sex, the proportion that fail to switch can vary but the processes are not separate. I don’t suggest that this actually happens but nor do I think the presented evidence confirms separate processes of gametocytogenesis. The author’s intended meaning is still somewhat opaque in the use of the word “terminal”. Gametocytogenesis is a terminal process, it is unclear whether terminal encompasses the entire process (as I assumed above) or merely the later stages when gametocyte sex can be established. If the latter, then the observation is facile that further male and female differentiation are separate processes after male and female sex has been established. If gametocytogenesis is proposed to be sexually divergent from initiation then there is some implicit difference that precedes or is concurrent with AP2G initiating commitment, is there any evidence for such a difference? I don’t think the authors provide any evidence of such a mechanism and I think this section should be redrafted much more cautiously and the actually intended meaning conveyed much more explicitly. If this section is just speculation then comments re “confirming” separate processes of gametocytogenesis should be removed.

My comments re the observed increase in H4 expression following Cq treatment and the comments on H3.3 were not meant to imply that the paper lacked a discussion of histone gene expression but rather to indicate that increased nucleosome density as a resistance mechanism was not the only possible explanation for altered H4 expression levels, increased H3.3-H4 deposition would alter the composition but not the density of nucleosomes. I don’t suggest that the authors need to address H3.3 but they have now modified the MS to include a fairly long section on general histone gene expression and the expression of several chromatin regulatory proteins during the Pv lifecycle as revealed by the scRNAseq. Is this novel and noteworthy? The transcriptional profile of Pf histone genes has been long known from bulk transcriptomes. 

The authors table s3 shows that nearly all of the sex-specific genes identified in Pv scRNAseq in this study were also found in Pb by Yeoh et al using bulk rnaseq of sex specific separated gams also many were shown by obaldia and marti. All of these papers should be cited in the main body of the MS and included in the text not just the supplement. The authors point out the majority of the female gam genes they describe are restricted to this stage and sex but many of these genes are present in male and female Pb gams according to table s3, this should be commented on, is this a genuine species difference or possibly a technical artefact?

---

## [Editor Report · Decision Letter 3]

18 Mar 2020

Dear Dr Serre,

Thank you for submitting your further revised Methods and Resources entitled "Single-cell transcription analysis of Plasmodium vivax blood-stage parasites identifies stage- and species-specific profiles of expression" for publication in PLOS Biology. I have now obtained advice from the Academic Editor who has assessed your revisions. 

Based on our Academic Editor's evaluation, we will probably accept this manuscript for publication only if you will include a previously requested, straightforward validation of your assignment of reads to the 3' ends of genes (see below). We don't think this will take long. Below are our Academic Editor's full comments which contain an additional minor request.

Comments from the Academic Editor:

The authors have largely addressed the queries raised by the reviewer and myself (editor). 

Fig s2 is difficult to find in the attached revised MS, it is in suppl materials but its presence there is not indicated in the main MS, the suppl materials file could be renamed to indicate that it contains many of the suppl figs. I assume the legend in the response and fig s2 uses 5’-end in error when the authors mean 3’-end? If so, then I think this data is convincing that approx. half the reads are mapping within 500 bp of the end of the gene. 

The authors also show convincing examples of incorrect gene annotations which could explain the distant locations of their scRNAseq reads from the annotated gene 3’ ends. 

However, the authors don’t perform the requested broad comparison with bulk rnaseq because; “implementing a systematic reannotation and validation of gene isoforms in the P. vivax genome is going to be a major effort beyond the scope of the present report.” Maybe the authors have misunderstood the requested, trivial comparison to bulk RNAseq. The intention of the requested analysis was to distil multiple comments of R2 to a single validation of the assignment of reads to transcripts and genes. This could be addressed by comparing a table of bulk RNAseq assembled transcripts, eg from cufflinks, to the scRNAseq. This would not require any comparison with existing annotations. A simple bedtools intersect of the concatenated scRNAseq consensus peak coordinates and the bulk RNAseq transcripts should reveal the proportion of scRNAseq “peaks” that are corroborated by bulk RNAseq transcripts. The authors already have the bam files for the bulk rnaseq so this comparison would be trivial and would allay concerns that the scRNAseq was incorrectly assigning reads to distant genes.

I understood the reviewer to have agreed that the PCA plot shows that the variation in transcriptome of the approx. 5000 genes through development is largely concordant hence the similar eigenvectors resulting in a similar PCA. However, there will be some genes that are differently expressed between species across development. These are the genes that I understood the reviewer wanted discussed. However, upon re-reading the MS I think this additional discussion is probably unnecessary.

The authors have satisfactorily addressed all of the other comments and queries from the editor and reviewer.

We expect to receive your revised manuscript within two weeks. Your revisions should address the specific points made by each reviewer. In addition to the remaining revisions and before we will be able to formally accept your manuscript and consider it "in press", we also need to ensure that your article conforms to our guidelines. A member of our team will be in touch shortly with a set of requests. As we can't proceed until these requirements are met, your swift response will help prevent delays to publication.

*Copyediting*

*Published Peer Review History*

*Early Version*

*Submitting Your Revision*

Sincerely,

Di Jiang

PLOS Biology

ETHICS STATEMENT:

-- Please create a separate subsection entitled "Ethics Statement" and place it in the beginning of the Methods section. 

-- Please include the full name of the IACUC/ethics committee that reviewed and approved the animal care and use protocol/permit/project license. ***IMPORTANT: Please also include an approval number.***

-- Please include the specific national or international regulations/guidelines to which your animal care and use protocol adhered. Please note that institutional or accreditation organization guidelines (such as AAALAC) do not meet this requirement.

-- Please include information about the form of consent (written/oral) given for research involving human participants. All research involving human participants must have been approved by the authors' Institutional Review Board (IRB) or an equivalent committee, and all clinical investigation must have been conducted according to the principles expressed in the Declaration of Helsinki.

DATA POLICY:

-- Regardless of the method selected, please ensure that you provide the individual numerical values that underlie the summary data displayed in the following figure panels as they are essential for readers to assess your analysis and to reproduce it: Figures 1AB, 2AB, 3AB, 4AB, 5, S1AB, S2, S3, S4, S5. NOTE: the numerical data provided should include all replicates AND the way in which the plotted mean and errors were derived (it should not present only the mean/average values).

-- Please provide an editor/reviewer key/token to your data (sequence reads) deposited in NCBI SRA under the Bioproject PRJNA603327 so we can check it before accepting the manuscript for publication. 

---

## [Editor Report · Decision Letter 4]

15 Apr 2020

Dear Dr Serre,

On behalf of my colleagues and the Academic Editor, Michael Duffy, I am pleased to inform you that we will be delighted to publish your Methods and Resources in PLOS Biology. 

Early Version

PRESS 

Kind regards,

Vita Usova

Publishing Editor, 

PLOS Biology

on behalf of

Di Jiang,

Associate Editor

PLOS Biology